# Explicit Second-Order Min-Max Optimization: Practical Algorithms and Complexity Analysis

**Tianyi Lin**                                                                *tl3335@columbia.edu*
*Department of Industrial Engineering and Operations Research*
*Columbia University*

**Panayotis Mertikopoulos**                              *panayotis.mertikopoulos@imag.fr*
*Univ. Grenoble Alpes, CNRS, Inria, Grenoble INP, LIG, 38000 Grenoble, France*

**Michael I. Jordan**                                                   *jordan@cs.berkeley.edu*
*Departments of Electrical Engineering and Computer Sciences and Statistics*
*University of California, Berkeley*
*Inria Paris*

**Reviewed on OpenReview:** *https://openreview.net/forum?id=Hyk1GhEXGa*

## Abstract

We propose and analyze several inexact regularized Newton-type methods for finding a global saddle point of *convex-concave* unconstrained min-max optimization problems. Compared to first-order methods, our understanding of second-order methods for min-max optimization is relatively limited, as obtaining global rates of convergence with second-order information can be much more involved. In this paper, we examine how second-order information is used to speed up extra-gradient methods, even under inexactness. In particular, we show that the proposed methods generate iterates that remain within a bounded set and that the averaged iterates converge to an $\epsilon$-saddle point within $O(\epsilon^{-2/3})$ iterations in terms of a restricted gap function. We also provide a simple routine for solving the subproblem at each iteration, requiring a single Schur decomposition and $O(\log\log(1/\epsilon))$ calls to a linear system solver in a quasi-upper-triangular system. Thus, our method improves the existing line-search-based second-order min-max optimization methods (Monteiro & Svaiter, 2012; Bullins & Lai, 2022; Jiang & Mokhtari, 2025) by shaving off an $O(\log\log(1/\epsilon))$ factor in the required number of Schur decompositions. Finally, we evaluate our method on both synthetic benchmarks and a real-world application arising from AUC maximization on standard LIBSVM datasets, and find that the proposed second-order approach delivers stronger practical efficiency than representative first-order methods on these problems.

## 1 Introduction

Let $\mathbb{R}^m$ and $\mathbb{R}^n$ be finite-dimensional Euclidean spaces and assume that the function $f : \mathbb{R}^m \times \mathbb{R}^n \mapsto \mathbb{R}$ has a bounded and Lipschitz-continuous Hessian. We consider the following min-max optimization problem:

$$\min_{\mathbf{x}\in\mathbb{R}^m} \max_{\mathbf{y}\in\mathbb{R}^n} \; f(\mathbf{x},\mathbf{y}), \qquad (1.1)$$

and aim to develop methods for finding a global saddle point; i.e., a tuple $(\mathbf{x}^\star, \mathbf{y}^\star) \in \mathbb{R}^m \times \mathbb{R}^n$ such that

$$f(\mathbf{x}^\star, \mathbf{y}) \leq f(\mathbf{x}^\star, \mathbf{y}^\star) \leq f(\mathbf{x}, \mathbf{y}^\star), \quad \text{for all } \mathbf{x} \in \mathbb{R}^m, \; \mathbf{y} \in \mathbb{R}^n.$$

Throughout this paper, we will assume that the function $f(\mathbf{x},\mathbf{y})$ is convex in $\mathbf{x}$ for all $\mathbf{y} \in \mathbb{R}^n$ and concave in $\mathbf{y}$ for all $\mathbf{x} \in \mathbb{R}^m$. This *convex-concave* setting has been the focus of intense research in optimization, game

theory, economics and computer science for several decades (Von Neumann & Morgenstern, 1953; Dantzig, 1963; Blackwell & Girshick, 1979; Facchinei & Pang, 2007; Ben-Tal et al., 2009), and variants of this problem class have recently attracted significant interest in machine learning and data science, with applications to generative adversarial networks (GANs) (Goodfellow et al., 2014; Arjovsky et al., 2017), adversarial learning (Sinha et al., 2018), multi-agent systems (Shamma, 2008), and many other fields; see Facchinei & Pang (2007) and references therein for a wide range of concrete problems.

Motivated by these applications, several classes of optimization algorithms have been proposed for finding a global saddle point of Eq. (1.1) in the convex-concave setting. An important algorithm is the extragradient (EG) method (Korpelevich, 1976; Antipin, 1978; Nemirovski, 2004). The method's rate of convergence for smooth and strongly-convex-strongly-concave functions and bilinear functions (i.e., when $f(\mathbf{x}, \mathbf{y}) = \mathbf{x}^\top A\mathbf{y}$ for some square, full-rank matrix $A$) was shown to be linear by Korpelevich (1976) and Tseng (1995). Subsequently, Nemirovski (2004) showed that the method enjoys an $O(\epsilon^{-1})$ convergence guarantee for constrained problems with a bounded domain and a convex-concave function $f$. In unbounded domains, Solodov & Svaiter (1999) generalized EG to the hybrid proximal extragradient (HPE) method which provides a framework for analyzing the iteration complexity of several existing methods, including EG and Tseng's forward-backward splitting (Tseng, 2000), while Monteiro & Svaiter (2010) provided an $O(\epsilon^{-1})$ guarantee for HPE in both bounded and unbounded domains. In addition to EG, there are other methods that can achieve the same convergence guarantees, such as optimistic gradient descent ascent (OGDA) (Popov, 1980) and dual extrapolation (DE) (Nesterov, 2007); for a survey, see Hsieh et al. (2019) and the references therein. All methods are referred to as *optimal first-order methods* since they matched the lower bound (Ouyang & Xu, 2021).

Focusing on convex *minimization* problems for the moment, significant effort has been devoted to developing first-order methods that are characterized by simplicity of implementation and analytic tractability (Nesterov, 1983). In particular, it has been recognized that first-order methods are suitable for solving large-scale machine learning problems in which low-accuracy solutions may suffice (Sra et al., 2012; Lan, 2020). In contrast, second-order methods are known to enjoy superior convergence properties over their first-order counterparts in theory: the accelerated cubic regularized Newton method (Nesterov, 2008) and accelerated Newton proximal extragradient method (Monteiro & Svaiter, 2013) provably converge at a rate of $O(\epsilon^{-1/3})$ and $\tilde{O}(\epsilon^{-2/7})$ respectively, both exceeding the best possible $O(\epsilon^{-1/2})$ bound for first-order methods (Nemirovski & Yudin, 1983). Two optimal second-order methods with the rate of $O(\epsilon^{-2/7})$ have been recently proposed by Carmon et al. (2022) and Kovalev & Gasnikov (2022), independently. In practice, first-order methods may perform poorly in ill-conditioned problems and are known to be sensitive to the parameter choices in real-world applications in which second-order methods are recognized to be more robust and reliable (Pilanci & Wainwright, 2017; Roosta-Khorasani & Mahoney, 2019; Berahas et al., 2020).

In the context of convex-concave min-max problems, two issues arise: **(i)** achieving acceleration with second-order information is less tractable analytically; and **(ii)** acquiring accurate second-order information is computationally expensive in general. Aiming to address the first issue, a line of work has generalized practical first-order methods to their higher-order counterparts (Monteiro & Svaiter, 2012; Bullins & Lai, 2022; Jiang & Mokhtari, 2025), where the state-of-the-art iteration bound is $O(\epsilon^{-2/3} \log\log(1/\epsilon))$ (Jiang & Mokhtari, 2025). All these methods require a nontrivial *implicit* search scheme at every iteration, and this can be prohibitive in practice.[1] Another line of work has extended second-order convex optimization methods to their min-max counterparts (Nesterov, 2006; Huang et al., 2022; Chen et al., 2025a). Specifically, the first extension of the cubic regularized Newton method (Nesterov & Polyak, 2006) was provided in Nesterov (2006) and was shown to achieve a global convergence rate of $O(\epsilon^{-1})$. Huang et al. (2022) proposed a two-phase extension of cubic regularized Newton method (Nesterov & Polyak, 2006) and provided a theoretical guarantee under the error bound condition with a parameter $0 < \theta \leq 1$ (Huang et al., 2022, Assumption 5.1): the global rate is linear under a Lipschitz-type condition ($\theta = 1$) and $O(\epsilon^{-(1-\theta)/\theta^2})$ under a Hölder-type condition ($\theta \in (0,1)$). These conditions exclude important problem classes and are in general unverifiable.

Very recently, Chen et al. (2025a) established an improved upper bound of $\tilde{O}(\epsilon^{-4/7})$ iterations by extending the optimal second-order method of Carmon et al. (2022) to smooth convex-concave min-max problems.

---

[1]By "implicit," we mean that the method's inner-loop subproblem for computing the $k^{\text{th}}$ iterate involves the iterate being updated, leading to an implicit update rule. By contrast, "explicit" means that any inner-loop subproblem for computing the $k^{\text{th}}$ iterate does not involve the new iterate.

While such guarantee is theoretically appealing, the resulting procedure is a nested-loop algorithm and can be more involved to implement in practice. Our focus here is complementary: we develop *explicit*, practically implementable second-order methods whose core steps rely on standard numerical linear algebra primitives and remain well-motivated under the inexact and subsampled second-order information. In particular, each iteration reduces to solving structured linear systems (via a single Schur decomposition and a small number of quasi-triangular solves), and our subproblem solver requires only $O(\log \log(1/\epsilon))$ quasi-triangular linear solves to reach $\epsilon$-inexact solution of the subproblem. Empirically, this emphasis on explicitness yields strong practical performance on both synthetic benchmarks and real AUC maximization tasks on LIBSVM datasets.

Finally, it is worth mentioning that the existing second-order min-max optimization algorithms require exact second-order information; as a result, given the implicit nature of the inner loop subproblems involved, the methods' robustness to inexact information cannot be taken for granted. It is thus natural to ask:

> Can we develop explicit second-order min-max optimization algorithms that retain the same convergence guarantee even with inexact second-order information?

Our paper offers an affirmative answer to this question. Inspired by recent advances on variational inequalities (VIs) (Lin & Jordan, 2025), we start by presenting a "conceptual" second-order min-max optimization method with an iteration complexity of $O(\epsilon^{-2/3})$. Our convergence analysis closely follows that of Lin & Jordan (2025) although it is simpler in that it leverages the specific structure of unconstrained min-max optimization problems. Although our proof is not new, we include it for completeness.

We then propose a class of second-order min-max optimization methods that require only *inexact second-order information* and *inexact subproblem solutions*. Notably, the proposed inexact Jacobian regularity condition allows for using randomized sampling for solving finite-sum min-max problems. This yields considerable computational savings since the sample size increases gracefully from a very small sample set. We prove that our inexact methods can achieve an iteration complexity of $O(\epsilon^{-2/3})$ and the corresponding subsampled variants achieve the same bound with high probability. Our new subroutine for solving each subproblem involves a single Schur decomposition and $O(\log \log(1/\epsilon))$ calls to a linear system solver in a quasi-upper-triangular system. Thus, the total complexity bound is $O((m+n)^\omega \epsilon^{-2/3} + (m+n)^2 \epsilon^{-2/3} \log \log(1/\epsilon))$ where $\omega \approx 2.3728$ is the matrix multiplication constant (Demmel et al., 2007). In addition, we conduct experiments on synthetic and real data to demonstrate the efficiency of the proposed methods.

Our approach builds on existing algorithmic components: (i) the overall scheme is inspired by Lin & Jordan (2025); (ii) the subproblem reformulation is inspired by Adil et al. (2022); (iii) the use of the safeguarded Newton method and the corresponding iteration complexity bound of $O(\log \log(1/\epsilon))$ extend the analysis of Conn et al. (2000); Cartis et al. (2011); (iv) the subsampling techniques are inspired by Xu et al. (2020). However, we remark that combining these components to yield a theoretically sound and practically efficient algorithm is far from straightforward.

While Adil et al. (2022) showed how to formulate the subproblem as a root-finding problem, a central challenge remains: how to solve such problems efficiently. Techniques such as the generalized conjugate gradient method with Lanczos processes—which are effective in the minimization setting—do not directly extend to the min-max setting. In this context, Huang et al. (2022) proposed an alternative formulation (see Eq. (16) in their paper) and applied a damped Newton method. However, they did not provide theoretical guarantees, even in a local sense, due to the absence of key properties such as the non-singularity of the Jacobian matrix. Even when adopting the root-finding formulation from Adil et al. (2022), identifying a suitable solver is highly nontrivial. At first glance, bisection might seem preferable due to its global convergence guarantee, unlike Newton's method. One of our major contributions is to rigorously justify the use of the safeguarded Newton method by extending the analysis of Conn et al. (2000) and Cartis et al. (2011) from the minimization setting to the min-max setting. Moreover, it remains unclear if the existing inexactness conditions from Xu et al. (2020) can be directly combined with our overall scheme while sacrificing neither theoretical iteration complexity nor practical efficiency. For example, the proof of Lemma D.1 requires separate treatment of $|\Delta \mathbf{z}_{k-1}| \geq 1$ and $|\Delta \mathbf{z}_{k-1}| < 1$, which was unnecessary in previous works either due to stronger assumptions as in Lin & Jordan (2025) or the simpler analysis appropriate for a minimization setting as in Xu et al. (2020).

**Contribution.** Our goal is to develop practically effective second-order methods for min-max optimization without sacrificing worst-case iteration complexity. We propose a new class of algorithms that operates with *inexact* second-order information and *inexact* subproblem solutions while retaining the same theoretical guarantees. This relaxes requirements in prior work (Lin & Jordan, 2025; Adil et al., 2022), which either assume exact inner solves or do not characterize the cost of solving each subproblem. In terms of complexity, our method improves over the best available line-search approach in Jiang & Mokhtari (2025), which attains $O((m+n)^\omega \epsilon^{-2/3} \log\log(1/\epsilon))$. Subsequent work has introduced additional strategies that match or surpass our guarantees (Alves et al., 2024; Jiang et al., 2024; Chen et al., 2025b); incorporating these ideas into our framework is a natural direction, but we leave them to future work. Conceptually, our contribution is to *explicitize* and *safeguard* a Newton-type method under inexact Jacobians and inexact subproblem solves, yielding a markedly better dependence on the target accuracy while remaining implementable with standard numerical linear algebra primitives.

**Notations.** We use bold lower-case letters to denote vectors. For $f(\cdot) : \mathbb{R}^n \to \mathbb{R}$, we let $\nabla f(\mathbf{z})$ denote the gradient of $f$ at $\mathbf{z}$. For $f(\cdot, \cdot) : \mathbb{R}^m \times \mathbb{R}^n \to \mathbb{R}$, we let $\nabla_{\mathbf{x}} f(\mathbf{x}, \mathbf{y})$ or $\nabla_{\mathbf{y}} f(\mathbf{x}, \mathbf{y})$ denote the partial gradient of $f$ at $(\mathbf{x}, \mathbf{y})$. We use $\nabla f(\mathbf{x}, \mathbf{y})$ to denote the gradient at $(\mathbf{x}, \mathbf{y})$ where $\nabla f(\mathbf{x}, \mathbf{y}) = (\nabla_{\mathbf{x}} f(\mathbf{x}, \mathbf{y}), \nabla_{\mathbf{y}} f(\mathbf{x}, \mathbf{y}))$ and $\nabla^2 f(\mathbf{x}, \mathbf{y})$ to denote the Hessian at $(\mathbf{x}, \mathbf{y})$. Finally, we use $O(\cdot), \Omega(\cdot)$ to hide absolute constants which do not depend on problem parameters, and $\tilde{O}(\cdot), \tilde{\Omega}(\cdot)$ to hide absolute constants and additional log factors.

## 2 Preliminaries

We present the setup of min-max optimization under study, and we provide the definitions for functions as well as optimality criteria considered. In this regard, the regularity conditions that we impose for the function $f : \mathbb{R}^{m+n} \mapsto \mathbb{R}$ are as follows:

**Definition 2.1** *A function $f$ is $\rho$-Hessian Lipschitz if $\|\nabla^2 f(\mathbf{z}) - \nabla^2 f(\mathbf{z}')\| \leq \rho \|\mathbf{z} - \mathbf{z}'\|$ for $\forall \mathbf{z}, \mathbf{z}'$.*

**Definition 2.2** *A differentiable function $f$ is convex-concave if*

$$\begin{aligned}
f(\mathbf{x}', \mathbf{y}) &\geq f(\mathbf{x}, \mathbf{y}) + (\mathbf{x}' - \mathbf{x})^\top \nabla_{\mathbf{x}} f(\mathbf{x}, \mathbf{y}), \quad \text{for } \mathbf{x}', \mathbf{x} \in \mathbb{R}^m \text{ and any fixed } \mathbf{y} \in \mathbb{R}^n, \\
f(\mathbf{x}, \mathbf{y}') &\leq f(\mathbf{x}, \mathbf{y}) + (\mathbf{y}' - \mathbf{y})^\top \nabla_{\mathbf{y}} f(\mathbf{x}, \mathbf{y}), \quad \text{for } \mathbf{y}', \mathbf{y} \in \mathbb{R}^n \text{ and any fixed } \mathbf{x} \in \mathbb{R}^m.
\end{aligned}$$

We define the notion of global saddle points for the problem in Eq. (1.1).

**Definition 2.3** *A point $\mathbf{z}^\star = (\mathbf{x}^\star, \mathbf{y}^\star) \in \mathbb{R}^m \times \mathbb{R}^n$ is a global saddle point of a function $f(\cdot, \cdot)$ if $f(\mathbf{x}^\star, \mathbf{y}) \leq f(\mathbf{x}^\star, \mathbf{y}^\star) \leq f(\mathbf{x}, \mathbf{y}^\star)$ for all $\mathbf{x} \in \mathbb{R}^m$ and $\mathbf{y} \in \mathbb{R}^n$.*

Throughout this paper, we assume that the following conditions are satisfied.

**Assumption 2.4** *The function $f(\mathbf{x}, \mathbf{y})$ is continuously differentiable and convex-concave, and at least one global saddle point of $f(\mathbf{x}, \mathbf{y})$ exists.*

**Assumption 2.5** *The function $f(\mathbf{x}, \mathbf{y})$ is $\rho$-Hessian Lipschitz.*

The existence of a global saddle point $\mathbf{z}^\star = (\mathbf{x}^\star, \mathbf{y}^\star)$ under Assumption 2.4 guarantees that $f(\mathbf{x}^\star, \mathbf{y}) \leq f(\mathbf{x}^\star, \mathbf{y}^\star) \leq f(\mathbf{x}, \mathbf{y}^\star)$ for all $\mathbf{x} \in \mathbb{R}^m$ and $\mathbf{y} \in \mathbb{R}^n$. Thus, we adopt a restricted gap function (Nesterov, 2007) to provide a measure for the optimality of $\hat{\mathbf{z}} = (\hat{\mathbf{x}}, \hat{\mathbf{y}})$ in the unconstrained convex-concave setting[2].

**Definition 2.6** *The restricted gap function is defined by*

$$\text{gap}(\hat{\mathbf{z}}, \beta) = \max_{\mathbf{y}: \|\mathbf{y} - \mathbf{y}^\star\| \leq \beta} f(\hat{\mathbf{x}}, \mathbf{y}) - \min_{\mathbf{x}: \|\mathbf{x} - \mathbf{x}^\star\| \leq \beta} f(\mathbf{x}, \hat{\mathbf{y}})$$

*where $\beta$ satisfies that $\|\hat{\mathbf{z}} - \mathbf{z}^\star\| \leq \beta$. Clearly, we have $\text{gap}(\hat{\mathbf{z}}, \beta) \geq 0$ since $f(\mathbf{x}^\star, \hat{\mathbf{y}}) \leq f(\hat{\mathbf{x}}, \mathbf{y}^\star)$.*

---

[2]The restricted gap is also related to the classical Nikaidô-Isoda function Nikaidô & Isoda (1955) defined for a class of noncooperative convex games in a more general setting.

**Remark 2.7** *The restricted gap in Definition 2.6 is a theoretical quantity defined relative to a global saddle point $\mathbf{z}^\star$. In our experiments, we compute this metric in two standard ways depending on whether $\mathbf{z}^\star$ is available in closed form. For cubic regularized bilinear min-max problems, $\mathbf{z}^\star$ can be derived in closed form, so we compute the restricted gap exactly at every iterate. For AUC maximization problem, where a closed-form saddle point is unavailable, we follow the practice of replacing $\mathbf{z}^\star$ by a high-accuracy proxy obtained from a strong baseline run with a stringent stopping rule; in particular, we report the restricted gap relative to the best-found solution among all methods after a sufficiently long budget. This protocol makes the restricted gap operational while preserving its role as a stable and comparable progress certificate across solvers.*

**Definition 2.8** *A point $\hat{\mathbf{z}} = (\hat{\mathbf{x}}, \hat{\mathbf{y}})$ is an $\epsilon$-global saddle point of a convex-concave function $f(\cdot, \cdot)$ if $\mathrm{gap}(\hat{\mathbf{z}}, \beta) \leq \epsilon$. If $\epsilon = 0$, it is a global saddle point.*

In our method, we denote the $k^{\text{th}}$ iterate by $(\mathbf{x}_k, \mathbf{y}_k)$ and we define the averaged (ergodic) iterates by $(\bar{\mathbf{x}}_k, \bar{\mathbf{y}}_k)$. In particular, given a sequence of weights $\{\lambda_k\}_{k=1}^T$, we let

$$\bar{\mathbf{x}}_k = \frac{1}{\sum_{i=1}^k \lambda_i} \left( \sum_{i=1}^k \lambda_i \mathbf{x}_i \right), \quad \bar{\mathbf{y}}_k = \frac{1}{\sum_{i=1}^k \lambda_i} \left( \sum_{i=1}^k \lambda_i \mathbf{y}_i \right). \tag{2.1}$$

In our convergence analysis, we define $\mathbf{z} = (\mathbf{x}, \mathbf{y}) \in \mathbb{R}^{m+n}$ and the operator $F : \mathbb{R}^{m+n} \mapsto \mathbb{R}^{m+n}$:

$$F(\mathbf{z}) = \begin{bmatrix} \nabla_{\mathbf{x}} f(\mathbf{x}, \mathbf{y}) \\ -\nabla_{\mathbf{y}} f(\mathbf{x}, \mathbf{y}) \end{bmatrix}. \tag{2.2}$$

Accordingly, the Jacobian of $F$ is defined as follows (note that $DF$ is asymmetric in general),

$$DF(\mathbf{z}) = \begin{bmatrix} \nabla_{\mathbf{xx}}^2 f(\mathbf{x}, \mathbf{y}) & \nabla_{\mathbf{xy}}^2 f(\mathbf{x}, \mathbf{y}) \\ -\nabla_{\mathbf{xy}}^2 f(\mathbf{x}, \mathbf{y}) & -\nabla_{\mathbf{yy}}^2 f(\mathbf{x}, \mathbf{y}) \end{bmatrix} \in \mathbb{R}^{(m+n) \times (m+n)}. \tag{2.3}$$

In the following lemma, we provide the properties of $F$ and its Jacobian $DF$ under Assumptions 2.4 and 2.5.

**Lemma 2.9** *Suppose that Assumptions 2.4 and 2.5 hold. Then, we have that (a) $F$ is monotone, i.e., $(\mathbf{z} - \mathbf{z}')^\top (F(\mathbf{z}) - F(\mathbf{z}')) \geq 0$, (b) $DF$ is $\rho$-Lipschitz continuous, i.e., $\|DF(\mathbf{z}) - DF(\mathbf{z}')\| \leq \rho \|\mathbf{z} - \mathbf{z}'\|$, and (c) $F(\mathbf{z}^\star) = 0$ for any global saddle point $\mathbf{z}^\star \in \mathbb{R}^{m+n}$ of the function $f(\cdot, \cdot)$.*

Before proceeding to our algorithmic framework and convergence analysis, we present the following well-known result which will be used in the subsequent analysis. Given its importance, we provide the proof for completeness in the appendix.

**Proposition 2.10** *Let $(\bar{\mathbf{x}}_k, \bar{\mathbf{y}}_k)$ and $F(\cdot)$ be defined in Eq. (2.1) and (2.2). Then, under Assumption 2.4, the following statement holds true,*

$$f(\bar{\mathbf{x}}_k, \mathbf{y}) - f(\mathbf{x}, \bar{\mathbf{y}}_k) \leq \frac{1}{\sum_{i=1}^k \lambda_i} \left( \sum_{i=1}^k \lambda_i (\mathbf{z}_i - \mathbf{z})^\top F(\mathbf{z}_i) \right).$$

## 3 Conceptual Algorithm and Convergence Analysis

As a warm-up, we describe the scheme of Newton-MinMax which is a second-order version of the method in Lin & Jordan (2025) for min-max optimization and yields an iteration complexity of $\Theta(\epsilon^{-2/3})$. We emphasize that Newton-MinMax is a *conceptual* algorithmic framework since it requires exact second-order information and requires the cubic regularized subproblem to be solved exactly. See additional proofs in Appendix C.

### 3.1 Algorithmic scheme

We summarize our second-order method, which we call Newton-MinMax($\mathbf{z}_0$, $\rho$, $T$), in Algorithm 1 where $\mathbf{z}_0 = (\mathbf{x}_0, \mathbf{y}_0) \in \mathbb{R}^m \times \mathbb{R}^n$ is an initial point, $\rho > 0$ is a Lipschitz constant for the Hessian of the function $f$

---

**Algorithm 1** Newton-MinMax($\mathbf{z}_0$, $\rho$, $T$)

---

**Input:** initial point $\mathbf{z}_0$, Lipschitz parameter $\rho$ and iteration number $T \geq 1$.
**Initialization:** set $\hat{\mathbf{z}}_0 = \mathbf{z}_0$.
**for** $k = 0, 1, 2, \ldots, T-1$ **do**
    **STEP 1:** If $\mathbf{z}_k$ is a global saddle point of the problem in Eq. (1.1), then **stop**.
    **STEP 2:** Compute an *exact* solution $\Delta\mathbf{z}_k$ of the subproblem $F(\hat{\mathbf{z}}_k) + DF(\hat{\mathbf{z}}_k)\Delta\mathbf{z}_k + 6\rho\|\Delta\mathbf{z}_k\|\Delta\mathbf{z}_k = \mathbf{0}$.
    **STEP 3:** Compute $\lambda_{k+1} > 0$ such that $\frac{1}{33} \leq \lambda_{k+1}\rho\|\Delta\mathbf{z}_k\| \leq \frac{1}{13}$.
    **STEP 4:** Compute $\mathbf{z}_{k+1} = \hat{\mathbf{z}}_k + \Delta\mathbf{z}_k$.
    **STEP 5:** Compute $\hat{\mathbf{z}}_{k+1} = \hat{\mathbf{z}}_k - \lambda_{k+1}F(\mathbf{z}_{k+1})$.
**end for**
**Output:** $\bar{\mathbf{z}}_T = \frac{1}{\sum_{k=1}^{T}\lambda_k}\left(\sum_{k=1}^{T}\lambda_k\mathbf{z}_k\right)$.

---

and $T \geq 1$ is an iteration number. Our method is a generalization of the first-order extragradient method. Indeed, the $k^{\text{th}}$ iteration consists of two important algorithmic components: (1) we compute $\mathbf{z}_{k+1} = \hat{\mathbf{z}}_k + \Delta\mathbf{z}_k$ where $\Delta\mathbf{z}_k \in \mathbb{R}^m \times \mathbb{R}^n$ is an exact solution of the *nonlinear equation* problem given by $F(\hat{\mathbf{z}}_k) + DF(\hat{\mathbf{z}}_k)\Delta\mathbf{z}_k + 6\rho\|\Delta\mathbf{z}_k\|\Delta\mathbf{z}_k = \mathbf{0}$, and (2) we compute $\hat{\mathbf{z}}_{k+1} = \hat{\mathbf{z}}_k - \lambda_{k+1}F(\mathbf{z}_{k+1})$.

As suggested by Lin & Jordan (2025), we choose to update $\lambda_k$ in an adaptive manner and prove that our method can achieve an iteration complexity of $O(\epsilon^{-2/3})$ under Assumptions 2.4 and 2.5. Such a strategy makes sense; indeed, $\lambda_k$ is the step size and would be better to increase as the iterate $\mathbf{z}_k$ approaches the set of global saddle points where the value of $\|\Delta\mathbf{z}_k\|$ measures the closeness. From a practical viewpoint, Algorithm 1 serves as an alternative to the current pipeline of line-search-based methods – which it simplifies by removing the need for an implicit binary search.

**Theorem 3.1** *Suppose that Assumptions 2.4 and 2.5 hold. Then, the sequence of iterates generated by Algorithm 1 is bounded and, in addition*

$$\mathrm{gap}(\bar{\mathbf{z}}_T, \beta) \leq \frac{2112\sqrt{3}\rho\|\mathbf{z}_0 - \mathbf{z}^\star\|^3}{T^{3/2}},$$

*where $\mathbf{z}^\star = (\mathbf{x}^\star, \mathbf{y}^\star)$ is a global saddle point, $\rho > 0$ is defined as in Assumption 2.5, and $\beta = 7\|\mathbf{z}_0 - \mathbf{z}^\star\|$. As such, Algorithm 1 achieves an $\epsilon$-global saddle point solution within $O(\epsilon^{-2/3})$ iterations.*

Although Algorithm 1 is conceptual, it forms the basis for the material in the next section, where we relax the strong requirements of Algorithm 1 and propose a class of second-order min-max optimization methods that require only inexact second-order information and inexact subproblem solutions.

## 4 Inexact Algorithm and Complexity Analysis

Building on the conceptual algorithm of the previous section, we present the Inexact-Newton-MinMax scheme, and we provide a global convergence guarantee in terms of the number of iterations required until convergence. Our inexact Jacobian regularity condition is inspired by Xu et al. (2020) and it allows for the use of randomized sampling for solving finite-sum min-max optimization problems. Our subroutine, which is inspired by Adil et al. (2022), can solve each subproblem using a single Schur decomposition and $O(\log\log(1/\epsilon))$ calls to a linear system solver in a quasi-upper-triangular system. See additional proofs in Appendix D.

### 4.1 Algorithmic scheme

We summarize our inexact second-order method, which we call Inexact-Newton-MinMax($\mathbf{z}_0$, $\rho$, $T$), in Algorithm 2 where $\mathbf{z}_0$ is an initial point, $\rho > 0$ is a Lipschitz constant for the Hessian of the function $f$ and $T \geq 1$ is an iteration number.

Our method combines Algorithm 1 with an inexact second-order framework (Xu et al., 2020) in the context of min-max optimization. Indeed, the key difference is that Eq. (4.1) used the inexact Jacobian $J(\hat{\mathbf{z}}_k)$, which can be formed and evaluated efficiently in practice. Throughout this section, we impose the following two conditions on the inexact Jacobian construction and the inexact subproblem solving.

---

**Algorithm 2** Inexact-Newton-MinMax($\mathbf{z}_0$, $\rho$, $T$)

---

**Input:** initial point $\mathbf{z}_0$, Lipschitz parameter $\rho$ and iteration number $T \geq 1$.
**Initialization:** set $\hat{\mathbf{z}}_0 = \mathbf{z}_0$ as well as $\kappa_J > 0$, $0 < \kappa_m < \min\{1, \frac{\rho}{4}\}$ and $0 \leq \tau_0 < \frac{\rho}{4}$.
**for** $k = 0, 1, 2, \ldots, T-1$ **do**
    **STEP 1:** If $\mathbf{z}_k$ is a global saddle point of the problem in Eq. (1.1), then **stop**.
    **STEP 2:** Compute an *inexact* solution $\Delta\mathbf{z}_k$ of the subproblem

$$F(\hat{\mathbf{z}}_k) + J(\hat{\mathbf{z}}_k)\Delta\mathbf{z}_k + 6\rho\|\Delta\mathbf{z}_k\|\Delta\mathbf{z}_k = \mathbf{0}. \tag{4.1}$$

such that Condition 4.1 and 4.2 hold true with a proper choice of $\tau_k \geq 0$.
    **STEP 3:** Compute $\lambda_{k+1} > 0$ such that $\frac{1}{30} \leq \lambda_{k+1}\rho\|\Delta\mathbf{z}_k\| \leq \frac{1}{14}$.
    **STEP 4:** Compute $\mathbf{z}_{k+1} = \hat{\mathbf{z}}_k + \Delta\mathbf{z}_k$.
    **STEP 5:** Compute $\hat{\mathbf{z}}_{k+1} = \hat{\mathbf{z}}_k - \lambda_{k+1}F(\mathbf{z}_{k+1})$.
**end for**
**Output:** $\bar{\mathbf{z}}_T = \frac{1}{\sum_{k=1}^{T} \lambda_k} \left(\sum_{k=1}^{T} \lambda_k\mathbf{z}_k\right)$.

---

**Condition 4.1 (Inexact Jacobian regularity)** *For some $\kappa_J > 0$ and $\tau_k \geq 0$, the inexact Jacobian $J(\hat{\mathbf{z}}_k)$ satisfies the following regularity conditions: $\|(J(\hat{\mathbf{z}}_k) - DF(\hat{\mathbf{z}}_k))\Delta\mathbf{z}_k\| \leq \tau_k\|\Delta\mathbf{z}_k\|$ and $\|J(\hat{\mathbf{z}}_k)\| \leq \kappa_J$, where $\{\hat{\mathbf{z}}_k\}_{k\geq 0}$ and $\{\Delta\mathbf{z}_k\}_{k\geq 0}$ are generated by Algorithm 2.*

**Condition 4.2 (Sufficient inexact solving)** *Fixing $\kappa_m \in (0,1)$, we solve the nonlinear equation problem in Eq. (4.1) inexactly to find $\Delta\mathbf{z}_k$ such that $\|F(\hat{\mathbf{z}}_k)+J(\hat{\mathbf{z}}_k)\Delta\mathbf{z}_k+6\rho\|\Delta\mathbf{z}_k\|\Delta\mathbf{z}_k\| \leq \kappa_m \min\{\|\Delta\mathbf{z}_k\|^2, \|F(\hat{\mathbf{z}}_k)\|\}$. In addition, $\{\hat{\mathbf{z}}_k\}_{k\geq 0}$ and $\{\Delta\mathbf{z}_k\}_{k\geq 0}$ are generated by Algorithm 2.*

Under Conditions 4.1 and 4.2, our proposed algorithm (cf. Algorithm 2) achieves the same worst-case iteration complexity of $O(\epsilon^{-2/3})$ for computing an $\epsilon$-global saddle point of the problem in Eq. (1.1) as that of the exact variant (cf. Algorithm 1).

Notably, Condition 4.1 allows for the principled use of many practical techniques to constructing the inexact Jacobian $J(\hat{\mathbf{z}}_k)$ in the context of min-max optimization. One such scheme can be described for solving the *finite-sum* min-max optimization problems in the form of

$$\min_{\mathbf{x}\in\mathbb{R}^m} \max_{\mathbf{y}\in\mathbb{R}^n} \ f(\mathbf{x},\mathbf{y}) \triangleq \tfrac{1}{N}\sum_{i=1}^{N} f_i(\mathbf{x},\mathbf{y}), \tag{4.2}$$

and its special instantiation

$$\min_{\mathbf{x}\in\mathbb{R}^m} \max_{\mathbf{y}\in\mathbb{R}^n} \ f(\mathbf{x},\mathbf{y}) \triangleq \tfrac{1}{N}\sum_{i=1}^{N} f_i(\mathbf{a}_i^\top\mathbf{x}, \mathbf{b}_i^\top\mathbf{y}), \tag{4.3}$$

where $N \gg 1$, each of $f_i$ is a convex-concave function with bounded and $\rho$-Lipschitz Hessian, and $\{(\mathbf{a}_i, \mathbf{b}_i)\}_{i=1}^{N} \subseteq \mathbb{R}^m \times \mathbb{R}^n$ are a collection of data samples.

This type of finite-sum problem is standard in optimization and machine learning (Shalev-Shwartz & Ben-David, 2014; Roosta-Khorasani et al., 2014; Roosta-Khorasani & Mahoney, 2019). Owing to the finite-sum structure, both uniform and nonuniform subsampling can be implemented to satisfy Condition 4.1 with high probability. Under standard sampling conditions, we obtain the stronger bound $\|J(\hat{\mathbf{z}}_k) - DF(\hat{\mathbf{z}}_k)\| \leq \tau_k$, which directly implies the corresponding requirement in Condition 4.1. As a consequence, by Theorem 4.1, the resulting subsampled Newton method preserves the same iteration complexity of $O(\epsilon^{-2/3})$ for solving finite-sum convex-concave min-max problems; see Theorem 4.13 for details.

While Condition 4.1 focuses on how to construct and control the inexact Jacobian $J(\hat{\mathbf{z}}_k)$ (see Section 4.3), Section 4.2 aims at solving the inner subproblem for a fixed $J(\hat{\mathbf{z}}_k)$. In particular, we propose a safeguarded Newton-bisection scheme that returns an update satisfying Condition 4.2 with a single Schur decomposition and only $O(\log\log(1/\epsilon))$ solves of quasi-upper-triangular linear systems. This modular design has separated Jacobian approximation from subproblem accuracy, enabling the overall complexity analysis by combining the guarantees for Conditions 4.1 and 4.2.

We provide the iteration complexity of Algorithm 2 as follows,

**Theorem 4.1** *Suppose that Assumption 2.4 and 2.5 hold and*

$$0 \leq \tau_k \leq \min\{\tau_0, \tfrac{\rho(1-\kappa_m)}{4(\kappa_J+6\rho)}\|F(\hat{\mathbf{z}}_k)\|\}, \text{ for all } k \geq 0.$$

*Then, the iterates generated by Algorithm 2 are bounded and, in addition,*

$$\text{gap}(\bar{\mathbf{z}}_T, \beta) \leq \tfrac{2112\sqrt{3}\rho\|\mathbf{z}_0-\mathbf{z}^\star\|^3}{T^{3/2}},$$

*where $\mathbf{z}^\star = (\mathbf{x}^\star, \mathbf{y}^\star)$ is a global saddle point, $\rho > 0$ is defined in Assumption 2.5, and $\beta = 7\|\mathbf{z}_0 - \mathbf{z}^\star\|$. As such, Algorithm 2 achieves an $\epsilon$-global saddle point solution within $O(\epsilon^{-2/3})$ iterations.*

**Remark 4.2** *Theorem 4.1 demonstrates that Algorithm 2 achieves the same iteration complexity as Algorithm 1 regardless of inexact second-order information and inexact subproblem solving under Conditions 4.1 and 4.2. Unfortunately, we are unable to claim optimality of Algorithm 2 since existing lower bounds (Adil et al., 2022; Lin & Jordan, 2025) have been established under the assumption of an exact second-order oracle and exact subproblem solutions. When moving to inexact methods, the oracle needs to be redefined. We want to emphasize that there may not be a universal way to define the oracle in this context, particularly when considering matrix operations. For instance, quasi-Newton and subsampled Newton methods are both categorized as inexact Newton methods, yet they differ in how second-order information is approximated, which impacts how their oracles should be defined and, consequently, how lower bounds are characterized. Recent work has begun to address this issue. Specifically, Agafonov et al. (2024a) proposed an accelerated stochastic second-order method that is robust to both gradient and Hessian inexactness, and established a lower bound under a $\delta$-inexact Hessian condition: $\|(H(\mathbf{x}) - \nabla^2 f(\mathbf{x}))(\mathbf{x}' - \mathbf{x})\| \leq \delta\|\mathbf{x}' - \mathbf{x}\|$ for all $\mathbf{x}, \mathbf{x}'$. Proceeding a further step, Agafonov et al. (2024b) extended the above results to the VI setting by considering a $\delta$-inexact Jacobian condition: $\|(J(\mathbf{z}) - DF(\mathbf{z}))(\mathbf{z}' - \mathbf{z})\| \leq \delta\|\mathbf{z}' - \mathbf{z}\|$ for all $\mathbf{z}, \mathbf{z}'$. They proved a lower bound and also provided a thorough comparison with our method (see pages 3-4). Notably, their lower bound is not valid for our inexact method since we bound the Jacobian inexactness with $\tau_k > 0$ that decreases as $k \to +\infty$ and bounds the norm of Jacobian from above.*

**Remark 4.3** *All of our algorithms require an upper bound $\rho$ on the Hessian Lipschitz constant, which enters only through the explicit safeguarded second-order update. In practice, one can obtain a conservative bound from problem structure when available (e.g., bounded features and bounded iterates in finite-sum problems), and we observe that the method is typically robust to moderate over-estimation of $\rho$ because the step selection is safeguarded. When such a bound is unavailable, a standard remedy is a backtracking scheme: start from an initial guess $\rho_0$ and increase it geometrically until the local model inequality used by the safeguard is satisfied, after which the iteration proceeds. We leave the development of a fully parameter-free variant with the same worst-case guarantees (i.e., jointly controlling the safeguard and the inexactness schedule) to future work, but emphasize that estimating smoothness parameters via backtracking is standard in large-scale optimization and does not significantly change the implementation of our scheme.*

## 4.2 Inexact subproblem solving and complexity analysis

We clarify how to obtain an inexact solution of the nonlinear equation problem in Eq. (4.1) such that Condition 4.2 holds. Indeed, we present a new safeguarded Newton-bisection scheme that solves each subproblem using a single Schur decomposition and $O(\log\log(1/\epsilon))$ calls to a linear system solver in a quasi-upper-triangular system. This yields a complexity of $O((m + n)^\omega \epsilon^{-2/3} + (m + n)^2 \epsilon^{-2/3}(\log\log(1/\epsilon)))$, which improves the practical performance over bisection and damped Newton while retaining a global convergence guarantee. In the case where $J(\hat{\mathbf{z}})$ is antisymmetric, we show that the Newton method is sufficient and requires $O(\log\log(1/\epsilon))$ calls to a linear system solver.

For simplicity, we rewrite the nonlinear equation problem in Eq. (4.1) as

$$F(\hat{\mathbf{z}}) + J(\hat{\mathbf{z}})\Delta\mathbf{z} + 6\rho\|\Delta\mathbf{z}\|\Delta\mathbf{z} = \mathbf{0}.$$

This is equivalent to finding a pair $(\Delta\mathbf{z}, \lambda)$ for which

$$(J(\hat{\mathbf{z}}) + \lambda I)\Delta\mathbf{z} = -F(\hat{\mathbf{z}}), \quad \lambda = 6\rho\|\Delta\mathbf{z}\|. \tag{4.4}$$

Although $J(\hat{\mathbf{z}})$ is not symmetric, it has a Schur decomposition $J(\hat{\mathbf{z}}) = QUQ^{-1}$ where $U$ is quasi-upper-triangular (since all entries of $J(\hat{\mathbf{z}})$ are real) in the sense that it is a block diagonal matrix with size $2 \times 2$ and $Q$ is a unitary matrix. This implies $J(\hat{\mathbf{z}}) + \lambda I = Q(U + \lambda I)Q^{-1}$. In the convex-concave setting, $J(\hat{\mathbf{z}})$ is monotone such that $v^\top J(\hat{\mathbf{z}})v \geq 0$ for all $v$, and $J(\hat{\mathbf{z}}) + \lambda I$ is invertible for all $\lambda > 0$.

In this regard, we define

$$\Delta\mathbf{z}(\lambda) \triangleq -(J(\hat{\mathbf{z}}) + \lambda I)^{-1}F(\hat{\mathbf{z}}) = -Q(U + \lambda I)^{-1}Q^{-1}F(\hat{\mathbf{z}}), \tag{4.5}$$

and obtain from Eq. (4.4) and the above definition that the solution $(\Delta\mathbf{z}, \lambda)$ we are looking for depends upon the nonlinear equality $\lambda = 6\rho\|\Delta\mathbf{z}(\lambda)\|$. For convenience, we define $\psi(\lambda) = \|\Delta\mathbf{z}(\lambda)\|^2$ and examine $\psi(\lambda)$ in the following proposition.

**Proposition 4.4** *The first-order and second-order derivatives of $\psi$ are given by $\psi'(\lambda) = -2\Delta\mathbf{z}(\lambda)^\top(J(\hat{\mathbf{z}}) + \lambda I)^{-1}\Delta\mathbf{z}(\lambda)$, and $\psi''(\lambda) = 2\|(J(\hat{\mathbf{z}}) + \lambda I)^{-1}\Delta\mathbf{z}(\lambda)\|^2 + 4\Delta\mathbf{z}(\lambda)^\top(J(\hat{\mathbf{z}}) + \lambda I)^{-2}\Delta\mathbf{z}(\lambda)$. If $F(\hat{\mathbf{z}}) \neq \mathbf{0}$ and $v^\top J(\hat{\mathbf{z}})v \geq 0$ for all $v$, then $\psi(\lambda)$ is strictly decreasing on $(0, \infty)$.*

The nonlinear equality $\lambda = 6\rho\|\Delta\mathbf{z}(\lambda)\|$ is equivalent to $\lambda = 6\rho\sqrt{\psi(\lambda)}$, which can be reformulated as the following one-dimensional nonlinear equation problem:

$$\phi(\lambda) \triangleq \sqrt{\psi(\lambda)} - \frac{\lambda}{6\rho} = 0. \tag{4.6}$$

In the following proposition, we examine $\phi(\lambda)$ and prove several key properties.

**Proposition 4.5** *Suppose that $F(\hat{\mathbf{z}}) \neq \mathbf{0}$ and $v^\top J(\hat{\mathbf{z}})v \geq 0$ for all $v$. Then, $\phi(\lambda)$ is strictly decreasing on $(0, \infty)$. Its first-order derivative is*

$$\phi'(\lambda) = -\frac{\Delta\mathbf{z}(\lambda)^\top(J(\hat{\mathbf{z}}) + \lambda I)^{-1}\Delta\mathbf{z}(\lambda)}{\|\Delta\mathbf{z}(\lambda)\|} - \frac{1}{6\rho} < 0, \quad \text{for all } \lambda > 0.$$

*Consequently, Eq. (4.6) admits a unique solution $\lambda^\star > 0$.*

The required solution is the unique positive root to Eq. (4.6). Let $\lambda^0 > 0$ be given with $\phi(\lambda^0) > 0$, we first perform one Schur decomposition: $J(\hat{\mathbf{z}}) = QUQ^{-1}$. Then, a typical iteration of the Newton method for finding such root replaces the current iterate $\lambda^j > 0$ with the candidate $\tilde{\lambda}^{j+1}$ for which

$$\tilde{\lambda}^{j+1} = \lambda^j - \frac{\phi(\lambda^j)}{\phi'(\lambda^j)}, \tag{4.7}$$

where $\phi(\lambda^j)$ can be obtained by solving $\Delta\mathbf{z}(\lambda^j) = -Q(U + \lambda^j I)^{-1}Q^{-1}F(\hat{\mathbf{z}})$, and $\phi'(\lambda^j)$ can be obtained using Proposition 4.5 once $\Delta\mathbf{z}(\lambda^j)$ is available. Note that $\phi(\lambda^j)$ and $\phi'(\lambda^j)$ can be computed using two calls to a linear system solver in a quasi-upper-triangular system.

In terms of convergence, the Newton method is not globally convergent for the general asymmetric case. We can adopt a safeguarded Newton-bisection scheme: given an interval $[L^j, U^j]$ satisfying $\phi(L^j) \geq 0 \geq \phi(U^j)$ and a current iterate $\lambda^j \in [L^j, U^j]$, we compute $\tilde{\lambda}^{j+1}$ using (4.7). If $\tilde{\lambda}^{j+1} \in (L^j, U^j)$, we accept it and set $\lambda^{j+1} = \tilde{\lambda}^{j+1}$; otherwise we take a bisection step and set $\lambda^{j+1} = (L^j + U^j)/2$. We then update the bracket using the sign of $\phi(\lambda^{j+1})$.

**Theorem 4.6** *Suppose that $F(\hat{\mathbf{z}}) \neq \mathbf{0}$ and $v^\top J(\hat{\mathbf{z}})v \geq 0$ for all $v$. Let $\lambda^\star > 0$ be the unique solution to Eq. (4.6). Suppose that the safeguarded Newton-bisection method is initialized with $[L^0, U^0]$ such that $\phi(L^0) \geq 0 \geq \phi(U^0)$. Then, $\lambda^j \to \lambda^\star$ globally. There also exists a constant $r > 0$ such that: (i) after at most $O(\log((U^0 - L^0)/r))$ iterations, the iterates enter the local Newton region $|\lambda - \lambda^\star| \leq r$, and (ii) once inside this region, all Newton steps are accepted and the convergence is Q-quadratic, requiring $O(\log\log(1/\epsilon))$ additional iterations to achieve $|\lambda^j - \lambda^\star| \leq \epsilon$. As a consequence, the total number of iterations is $O(\log(1/r)) + O(\log\log(1/\epsilon))$ where $r > 0$ is independent of $\epsilon$.*

We see from Theorem 4.6 that the total number of iterations to achieve $|\phi(\lambda^j)| \leq \epsilon'$ is also $O(\log\log(1/\epsilon'))$. This scalar reformulation is not merely a computational convenience: it provides a verifiable mechanism to enforce the inexact-solve requirement in Condition 4.2. Indeed, we rewrite $\phi(\lambda)$ as

$$\phi(\lambda) = \|\Delta\mathbf{z}(\lambda)\| - \tfrac{\lambda}{6\rho}.$$

Then, we have

$$
\begin{aligned}
&\|F(\hat{\mathbf{z}}) + J(\hat{\mathbf{z}})\Delta\mathbf{z}(\lambda) + 6\rho\|\Delta\mathbf{z}(\lambda)\|\Delta\mathbf{z}(\lambda)\| \\
&= \|(F(\hat{\mathbf{z}}) + (J(\hat{\mathbf{z}}) + \lambda I)\Delta\mathbf{z}(\lambda)) + (6\rho\|\Delta\mathbf{z}(\lambda)\| - \lambda)\Delta\mathbf{z}(\lambda)\| \\
&= |6\rho\|\Delta\mathbf{z}(\lambda)\| - \lambda|\|\Delta\mathbf{z}(\lambda)\| \\
&= 6\rho|\phi(\lambda)|\|\Delta\mathbf{z}(\lambda)\|.
\end{aligned}
$$

In addition, we rewrite Condition 4.2 as

$$\|F(\hat{\mathbf{z}}) + J(\hat{\mathbf{z}})\Delta\mathbf{z}(\lambda) + 6\rho\|\Delta\mathbf{z}(\lambda)\|\Delta\mathbf{z}(\lambda)\| \leq \kappa_m \min\{\|\Delta\mathbf{z}(\lambda)\|^2, \|F(\hat{\mathbf{z}})\|\}.$$

Thus, it suffices to guarantee

$$|\phi(\lambda)| \leq \tfrac{\kappa_m}{6\rho}\min\left\{\|\Delta\mathbf{z}(\lambda)\|, \tfrac{\|F(\hat{\mathbf{z}})\|}{\|\Delta\mathbf{z}(\lambda)\|}\right\},$$

which implies driving $|\phi(\lambda)|$ below the explicit tolerance threshold immediately yields the residual bound required by Condition 4.2. Consequently, the convergence $\lambda^j \to \lambda^\star$ delivered by our safeguarded Newton-bisection procedure implies that $\Delta\mathbf{z}(\lambda^j)$ satisfies Condition 4.2 after finitely many inner iterations, with no additional assumptions. Note that Condition 4.1 is enforced by the Jacobian construction (see Section 4.3) and is independent of $\{\lambda^j\}$; indeed, the role of $\{\lambda^j\}$ is to verify Condition 4.2 via the bound on $|\phi(\lambda)|$.

We consider the special case where $J(\hat{\mathbf{z}})^\top = -J(\hat{\mathbf{z}})$. By exploiting such special structure, we have $S \triangleq J(\hat{\mathbf{z}})^\top J(\hat{\mathbf{z}}) \succeq 0$ and $\psi(\lambda) = \|\Delta\mathbf{z}(\lambda)\|^2 = F(\hat{\mathbf{z}})^\top(\lambda^2 I + S)^{-1}F(\hat{\mathbf{z}})$. We let $\eta = \lambda^2$ and define

$$\Psi(\eta) \triangleq F(\hat{\mathbf{z}})^\top(\eta I + S)^{-1}F(\hat{\mathbf{z}}), \quad \Phi(\eta) \triangleq \Psi(\eta) - \tfrac{\eta}{36\rho^2}.$$

Then, solving $\phi(\lambda) = 0$ is equivalent to solving $\Phi(\eta) = 0$ with $\lambda^\star = \sqrt{\eta^\star}$.

**Proposition 4.7** *Suppose that $F(\hat{\mathbf{z}}) \neq \mathbf{0}$ and $J(\hat{\mathbf{z}})^\top = -J(\hat{\mathbf{z}})$. Then, $\Phi(\eta)$ is strictly decreasing and convex on $(0, \infty)$. Thus, $\Phi(\eta) = 0$ admits a unique solution $\eta^\star > 0$.*

**Theorem 4.8** *Suppose that $F(\hat{\mathbf{z}}) \neq \mathbf{0}$ and $J(\hat{\mathbf{z}})^\top = -J(\hat{\mathbf{z}})$. Let $\{\eta^j\}_{j\geq 0}$ be generated by the Newton iteration in the following form of*

$$\eta^{j+1} = \eta^j - \tfrac{\Phi(\eta^j)}{\Phi'(\eta^j)}, \quad \Phi(\eta^0) \geq 0.$$

*Then, $\eta^j$ increases monotonically and converges to the unique solution $\eta^\star$. Moreover, the rate is globally Q-linear with a factor at least $1 - \Phi'(\lambda^\star)/\Phi'(\lambda^0)$ and is ultimately Q-quadratic. Equivalently, we have $\lambda^j = \sqrt{\eta^j}$ converges to $\lambda^\star = \sqrt{\eta^\star}$, and the number of iterations to achieve $|\lambda^j - \lambda^\star| \leq \epsilon$ is $O(\log\log(1/\epsilon))$.*

Our reformulation closely follows Adil et al. (2022), who proposed a bisection-based solver and established the complexity bound of $O((m + n)^\omega \epsilon^{-2/3} + (m + n)^2 \epsilon^{-2/3}\log(1/\epsilon))$, where $\omega \approx 2.3728$ is the matrix-multiplication constant (Demmel et al., 2007). Our scheme improves the worst-case dependence on $\epsilon$ by leveraging a faster quadratic-convergence phase. Computationally, we perform a single Schur decomposition of an $(m+n)\times(m+n)$ matrix, which yields the $(m+n)^\omega$ arithmetic term and requires $O((m+n)^2)$ memory to store the factors. Consequently, the method is most suitable for moderate dimensions or structured instances (e.g., sparse, block-structured, or low-rank), while large-scale variants based on iterative Krylov solving or approximate factorization are natural but require additional analysis beyond the scope of this paper. Finally, once the decomposition is available, the remaining inner work depends only polylogarithmically on the target accuracy, leading to the $O(\log\log(1/\epsilon))$ dependence in the inexact subproblem solving.

Our analysis is related to Cartis et al. (2011, Theorem 6.3). The key difference lies in how asymmetry affects curvature properties of one-dimensional reformulations. In general asymmetric settings, neither concavity nor convexity of $\phi(\lambda)$ is guaranteed, and thus the Newton method requires safeguarding for global convergence. In contrast, under the antisymmetry condition, $\eta = \lambda^2$ restores convexity of the scalar equation and yields a globally convergent unmodified Newton scheme.

### 4.3 Finite-sum min-max optimization

We give concrete examples to clarify the ways to construct the inexact Jacobian such that Condition 4.1 holds true. The key ingredient is random sampling which can significantly reduce the computational cost in a minimization setting Xu et al. (2020) and we show that such technique can be employed for solving finite-sum min-max optimization problems in Eq. (4.2) and (4.3).

We let the probability distribution of sampling $\xi \in \{1, 2, \ldots, N\}$ be defined as $\text{Prob}(\xi = i) = p_i \geq 0$ for $i = 1, 2, \ldots, N$ and $\mathcal{S} \subseteq \{1, 2, \ldots, N\}$ denote a collection of sampled indices ($|\mathcal{S}|$ is its cardinality). Then, we can construct the inexact Jacobian as follows,

$$J(\mathbf{z}) = \frac{1}{N|\mathcal{S}|} \sum_{i \in \mathcal{S}} \frac{1}{p_i} DF_i(\mathbf{z}), \quad \text{for } DF_i(\mathbf{z}) = \begin{bmatrix} \nabla^2_{\mathbf{xx}} f_i(\mathbf{x}, \mathbf{y}) & \nabla^2_{\mathbf{xy}} f_i(\mathbf{x}, \mathbf{y}) \\ -\nabla^2_{\mathbf{xy}} f_i(\mathbf{x}, \mathbf{y}) & -\nabla^2_{\mathbf{yy}} f_i(\mathbf{x}, \mathbf{y}) \end{bmatrix}. \tag{4.8}$$

This construction is referred to as the subsampled Jacobian and can offer significant computational savings if $|\mathcal{S}| \ll N$ in big-data regime when $N \gg 1$.

In the general finite-sum setting with Eq. (4.2), we suppose

$$\sup_{\mathbf{z} \in \mathbb{R}^{m+n}} \|DF_i(\mathbf{z})\| \leq B_i, \quad \text{for all } i \in \{1, 2, \ldots, N\}, \tag{4.9}$$

and let $B_{\max} = \max_{1 \leq i \leq N} B_i$. The sample complexity results for the uniform sampling (i.e., $p_i = \frac{1}{N}$) are a consequence of Xu et al. (2020, Lemma 16) and we omit the proof for the brevity.

**Lemma 4.9** *Suppose that Eq.* (4.9) *holds true and let $B_{\max}$ and $0 < \tau, \delta < 1$ be defined properly. A uniform sampling with or without replacement is performed to form the subsampled Jacobian; indeed, $J(\mathbf{z})$ is constructed using Eq.* (4.8) *with $p_i = \frac{1}{n}$ and the sample size satisfies*

$$|\mathcal{S}| \geq \Theta^U(\tau, \delta) := \frac{16 B_{\max}^2}{\tau^2} \log\left(\frac{2(m+n)}{\delta}\right).$$

*Then, we have* $\text{Prob}(\|J(\mathbf{z}) - DF(\mathbf{z})\| \leq \tau) \geq 1 - \delta$.

**Remark 4.10** *Lemma 4.9 shows that the inexact Jacobian satisfies Condition 4.1 with probability $1 - \delta$ under certain choice of $\tau$ and $\kappa_J = B_{\max}$ if it is constructed using the uniform sampling and the size $|\mathcal{S}| = \Omega(\frac{B_{\max}^2}{\tau^2} \log(\frac{m+n}{\delta}))$. Indeed, the first inequality holds true with probability $1 - \delta$ since $\text{Prob}(\|J(\mathbf{z}) - DF(\mathbf{z})\| \leq \tau) \geq 1 - \delta$, and the second inequality holds true since $\kappa_H = B_{\max}$ (this is deterministic).*

In the finite-sum setting with Eq. (4.3), we construct a more "informative" distribution of sampling $\xi \in \{1, 2, \ldots, N\}$ as opposed to simplest uniform sampling. It is advantageous to bias the probability distribution towards carefully picking indices corresponding to those *relevant* $f_i$'s in forming the Jacobian. However, constructing inexact Hessian and corresponding sample complexity guarantee from Xu et al. (2020, Section 3.1) in a minimization setting requires $\nabla^2 f_i$ to be rank-one, which is not valid here. To address this, we avail ourselves of the operator-Bernstein inequality (Gross & Nesme, 2010).

We write $DF(\mathbf{z}) = \frac{1}{N} \sum_{i=1}^N \Lambda_i DF_i(\mathbf{a}_i^\top \mathbf{x}, \mathbf{b}_i^\top \mathbf{y}) \Lambda_i^\top$ where $\Lambda_i = \begin{bmatrix} \mathbf{a}_i & \\ & \mathbf{b}_i \end{bmatrix} \in \mathbb{R}^{(m+n) \times 2}$ and $DF_i(x, y) \in \mathbb{R}^{2 \times 2}$.

Then, the resulting compact form is $DF(\mathbf{z}) = \Lambda^\top \Sigma \Lambda$ where

$$\Lambda^\top = \begin{bmatrix} | & \cdots & | \\ \Lambda_1 & \cdots & \Lambda_N \\ | & \cdots & | \end{bmatrix} \text{ and } \Sigma = \frac{1}{N} \begin{bmatrix} DF_1(\mathbf{a}_i^\top \mathbf{x}, \mathbf{b}_i^\top \mathbf{y}) & \cdots & \cdots \\ \vdots & \ddots & \vdots \\ \cdots & \cdots & DF_N(\mathbf{a}_i^\top \mathbf{x}, \mathbf{b}_i^\top \mathbf{y}) \end{bmatrix}. \tag{4.10}$$

We assume that $B_{\text{avg}} = \frac{1}{N} \sum_{i=1}^N B_i$ and

$$\sup_{(\mathbf{x}, \mathbf{y})} \|DF_i(\mathbf{a}_i^\top \mathbf{x}, \mathbf{b}_i^\top \mathbf{y})\|(\|\mathbf{a}_i\|^2 + \|\mathbf{b}_i\|^2) \leq B_i, \quad \text{for all } i \in \{1, 2, \ldots, N\}. \tag{4.11}$$

---

**Algorithm 3** Subsampled-Newton-MinMax($\mathbf{z}_0$, $\rho$, $T$, $\delta$)

---

**Input:** initial point $\mathbf{z}_0$, Lipschitz parameter $\rho$, iteration number $T \geq 1$ and failure probability $\delta \in (0, 1)$.

**Initialization:** set $\hat{\mathbf{z}}_0 = \mathbf{z}_0$ as well as $0 < \kappa_m < \min\{1, \frac{\rho}{4}\}$ and $0 < \tau_0 < \frac{\rho}{4}$.

**for** $k = 0, 1, 2, \ldots, T-1$ **do**

**STEP 1:** If $\mathbf{z}_k$ is a global saddle point of the problem in Eq. (4.2) or (4.3), then **stop**.

**STEP 2:** Construct the inexact Jacobian $J(\hat{\mathbf{z}}_k)$ using Eq. (4.8) with the sample set of $|\mathcal{S}| \geq \Theta^U(\tau_k, 1 - \sqrt[T]{1-\delta})$ (*uniform*) or $|\mathcal{S}| \geq \Theta^N(\tau_k, 1 - \sqrt[T]{1-\delta})$ (*non-uniform*) given $0 < \tau_k \leq \min\{\tau_0, \frac{\rho(1-\kappa_m)}{4(B_{\max}+6\rho)}\|F(\hat{\mathbf{z}}_k)\|\}$.

**STEP 2:** Compute an *inexact* solution $\Delta\mathbf{z}_k$ of the following subproblem $F(\hat{\mathbf{z}}_k) + J(\hat{\mathbf{z}}_k)\Delta\mathbf{z}_k + 6\rho\|\Delta\mathbf{z}_k\|\Delta\mathbf{z}_k = \mathbf{0}$ such that Condition 4.2 hold true.

**STEP 3:** Compute $\lambda_{k+1} > 0$ such that $\frac{1}{30} \leq \lambda_{k+1}\rho\|\Delta\mathbf{z}_k\| \leq \frac{1}{14}$.

**STEP 4:** Compute $\mathbf{z}_{k+1} = \hat{\mathbf{z}}_k + \Delta\mathbf{z}_k$.

**STEP 5:** Compute $\hat{\mathbf{z}}_{k+1} = \hat{\mathbf{z}}_k - \lambda_{k+1}F(\mathbf{z}_{k+1})$.

**end for**

**Output:** $\bar{\mathbf{z}}_T = \frac{1}{\sum_{k=1}^T \lambda_k} \left(\sum_{k=1}^T \lambda_k \mathbf{z}_k\right)$.

---

For the uniform sampling with the particular nonuniform distribution given by

$$p_i = \frac{\|DF_i(\mathbf{a}_i^\top\mathbf{x}, \mathbf{b}_i^\top\mathbf{y})\|(\|\mathbf{a}_i\|^2 + \|\mathbf{b}_i\|^2)}{\sum_{i=1}^N \|DF_i(\mathbf{a}_i^\top\mathbf{x}, \mathbf{b}_i^\top\mathbf{y})\|(\|\mathbf{a}_i\|^2 + \|\mathbf{b}_i\|^2)}, \tag{4.12}$$

the following lemma summarizes the results on the sample complexity.

**Lemma 4.11** *Suppose that Eq. (4.11) holds and let $B_{\text{avg}}$ and $0 < \tau, \delta < 1$ be defined properly. Then, the nonuniform sampling is performed to form the subsampled Jacobian; indeed, $J(\mathbf{z})$ is constructed using Eq. (4.8) with $p_i > 0$ in Eq. (4.12) and the sample size satisfies*

$$|\mathcal{S}| \geq \Theta^N(\tau, \delta) := \frac{4B_{\text{avg}}^2}{\tau^2} \log\left(\frac{2(m+n)}{\delta}\right).$$

*Then, we have* $\text{Prob}(\|J(\mathbf{z}) - DF(\mathbf{z})\| \leq \tau) \geq 1 - \delta$.

**Remark 4.12** *Compared to Lemma 4.9, the computation of sampling probability in Lemma 4.11 requires going through the whole dataset and the cost is $O((m+n)N)$. Nonetheless, the computational savings with smaller sample size dominates such extra cost of computing the sampling probability in a minimization setting (Xu et al., 2016). In particular, the sample size from Lemma 4.11 is smaller as $B_{\text{avg}} \ll B_{\max}$ which occurs if one $B_i$ is much larger than the others. In addition, the sample size is proportional to the log of the failure probability in Lemma 4.9 and 4.11, allowing the use of a very small failure probability without increasing the sample size significantly.*

Combining Algorithm 2 and these random sampling strategies gives the first class of subsampled Newton methods for solving finite-sum min-max optimization problems. We summarize the scheme in Algorithm 3 and present the iteration complexity and total complexity with high-probability in the following theorem.

**Theorem 4.13** *Suppose that Assumptions 2.4 and 2.5 hold. Then, the iterates generated by Algorithm 3 are bounded and Algorithm 3 achieves an $\epsilon$-global saddle point solution within $O(\epsilon^{-2/3})$ iterations with the probability at least $1 - \delta$. As a consequence, the high-probability complexity bound is $O((m+n)^\omega \epsilon^{-2/3} + (m+n)^2\epsilon^{-2/3}\log\log(1/\epsilon))$ where $\omega \approx 2.3728$ is the multiplication constant.*

**Remark 4.14** *In finite-sum optimization, Arjevani & Shamir (2017) investigated lower bounds for second-order methods, where the complexity depends on the number of samples, the condition number, and the desired tolerance. They identified a gap between upper and lower bounds for subsampled Newton methods, indicating that simple sampling strategies are insufficient and that variance reduction or averaging could be essential for further improvement. Recent advances in stochastic second-order optimization methods with variance reduction further indicate the importance of combining structure and noise control (Zhou et al., 2019; Na et al., 2023). However, the lower bound on the oracle complexity of second-order methods for finite-sum minimax optimization remains an open question. This is beyond the scope of our current work but represents an important direction for future research.*

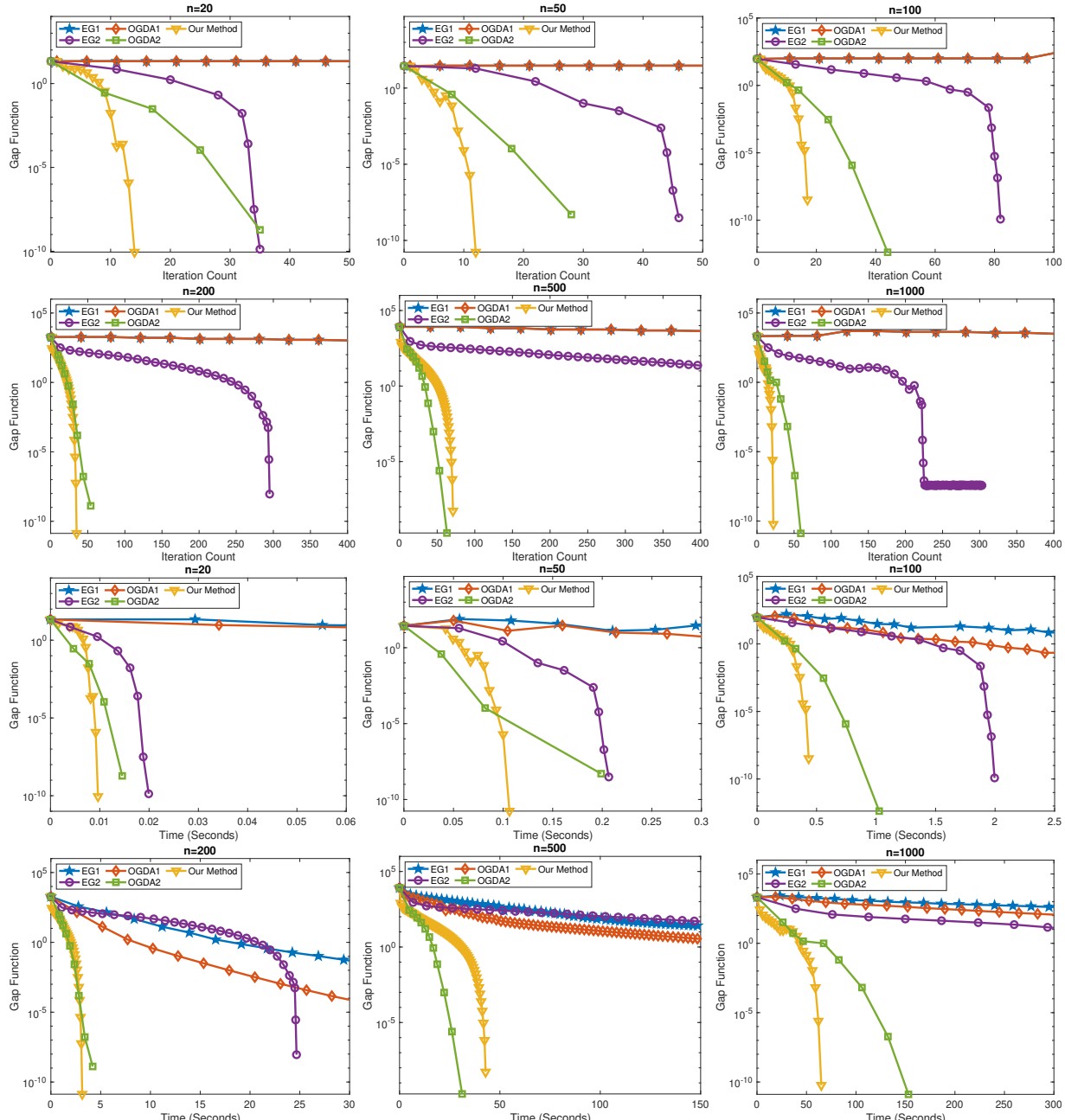

Figure 1: Performance of all algorithms for $n \in \{20, 50, 100, 200, 500, 1000\}$ when $\rho = \frac{1}{20n}$. The numerical results are presented in terms of iteration count (Top) and computational time (Bottom).

## 5 Numerical Experiments

We evaluate the performance of our methods for min-max optimization problems with synthetic and real data. The baseline methods include EG and OGDA (we refer to them as EG1 and OGDA1), their stochastic variants with or without variance reduction, and second-order variants of EG and OGDA (which we refer to as EG2 and OGDA2). We exclude the methods in Huang et al. (2022) since their convergence guarantees are proved under an error bound conditions. All the methods were implemented using MATLAB R2024b on a MacBook Pro with an Intel Core i9 2.4GHz and 16GB memory.

**Cubic regularized bilinear min-max problem.** Following the setup of Jiang & Mokhtari (2025), we consider the problem in the following form:

$$\min_{\mathbf{x}\in\mathbb{R}^n} \max_{\mathbf{y}\in\mathbb{R}^n} \ f(\mathbf{x},\mathbf{y}) = \tfrac{\rho}{6}\|\mathbf{x}\|^3 + \mathbf{y}^\top(A\mathbf{x}-\mathbf{b}), \tag{5.1}$$

where $\rho > 0$, the entries of $\mathbf{b} \in \mathbb{R}^n$ are generated independently from $[-1,1]$ and

$$A = \begin{bmatrix} 1 & -1 & & & \\ & 1 & -1 & & \\ & & \ddots & \ddots & \\ & & & 1 & -1 \\ & & & & 1 \end{bmatrix} \in \mathbb{R}^{n\times n}. \tag{5.2}$$

This min-max optimization problem is convex-concave and the function $f$ is $\rho$-Hessian Lipschitz. It has a unique global saddle point $\mathbf{x}^\star = A^{-1}\mathbf{b}$ and $\mathbf{y}^\star = -\tfrac{\rho}{2}\|\mathbf{x}^\star\|(A^\top)^{-1}\mathbf{x}^\star$. We use the restricted gap function defined in Section 2 as the evaluation metric. In our experiment, the parameters are chosen as $\rho = \tfrac{1}{20n}$ and $n \in \{50, 100, 200\}$. Since exact second-order information is available here, we use Algorithm 2 with $\tau_0 = 0$ (the exact Hessian) and compute each subproblem with $\kappa_m = 10^{-6}$ using our subroutine. The baseline methods include EG1, OGDA1, EG2 and OGDA2 where all of these methods require exact second-order information. We implement EG2 using the pseudocode of Bullins & Lai (2022, Algorithm 5.2 and 5.3) with the fine-tuning parameters. The implementation of OGDA2 is based on the code provided by the author of Jiang & Mokhtari (2025) with the line-search parameter $(\alpha,\beta) = (0.5, 0.8)$.

Figure 1 illustrates a clear accuracy gap between first- and second-order methods: the first-order baselines make little progress at stringent tolerances, whereas second-order methods reach high-accuracy solutions. Among the latter, our method is consistently competitive – and often faster in both iterations and wall-clock time – which we attribute to its safeguarded update without line search. This does not negate the value of line search in min-max optimization: the scheme of Jiang & Mokhtari (2025) can outperform our method with aggressive $(\alpha,\beta)$ on some instances, but such tuning is also less stable across runs and dimensions; we therefore use the conservative setting $(\alpha,\beta) = (0.5, 0.8)$ for robustness. The figure further reports results up to $n = 1000$, where our method remains stable as $n$ increases, though relative margins may narrow when conditioning and linear-algebra costs dominate. These results suggest that safeguards provide reliable time-to-accuracy, and it would be interesting future work to modify the line-search scheme of Jiang & Mokhtari (2025) to achieve similarly robust acceleration.

**AUC maximization problem.** The problem of maximizing an area under the receiver operating characteristic curve is a paradigm that learns a classifier for imbalanced data (Yang & Ying, 2022). The goal is to find a classifier $\theta \in \mathbb{R}^n$ that maximizes the AUC score on a set of samples $\{(\mathbf{a}_i, b_i)\}_{i=1}^N$, where $\mathbf{a}_i \in \mathbb{R}^n$ and $b_i \in \{-1, +1\}$. The min-max formulation for AUC maximization (Ying et al., 2016; Shen et al., 2018) is:

$$\min_{\mathbf{x}=(\theta,u,v)} \max_{y} \ \tfrac{1-\hat{p}}{N}\left\{\sum_{i=1}^N (\theta^\top\mathbf{a}_i - u)^2 \mathbb{I}_{[b_i=1]}\right\} + \tfrac{\hat{p}}{N}\left\{\sum_{i=1}^N (\theta^\top\mathbf{a}_i - v)^2 \mathbb{I}_{[b_i=-1]}\right\} \tag{5.3}$$

$$+ \tfrac{2(1+y)}{N}\left\{\sum_{i=1}^N \theta^\top\mathbf{a}_i(\hat{p}\mathbb{I}_{[b_i=-1]} - (1-\hat{p})\mathbb{I}_{[b_i=1]})\right\} + \tfrac{\rho}{6}\|\mathbf{x}\|^3 - \hat{p}(1-\hat{p})y^2,$$

where $\lambda > 0$ is a scalar, $\mathbb{I}_{[\cdot]}$ is an indicator function and $\hat{p} = \tfrac{\#\{i:b_i=1\}}{N}$ be the proportion of samples with positive label. It is clear that the min-max optimization problem in Eq. (5.3) is convex-concave and has the finite-sum structure in the form of Eq. (4.2) with the function $f_i(\mathbf{x}, y)$ given by

$$f_i(\mathbf{x},y) = (1-\hat{p})(\theta^\top\mathbf{a}_i - u)^2\mathbb{I}_{[b_i=1]} + \hat{p}(\theta^\top\mathbf{a}_i - v)^2\mathbb{I}_{[b_i=-1]}$$
$$+ 2(1+y)\theta^\top\mathbf{a}_i(\hat{p}\mathbb{I}_{[b_i=-1]} - (1-\hat{p})\mathbb{I}_{[b_i=1]}) + \tfrac{\rho}{6}\|\mathbf{x}\|^3 - \hat{p}(1-\hat{p})y^2.$$

The above function is in the cubic form and the min-max optimization problem has a global saddle point. We use the restricted gap function as the evaluation metric. In our experiment, the parameter is chosen

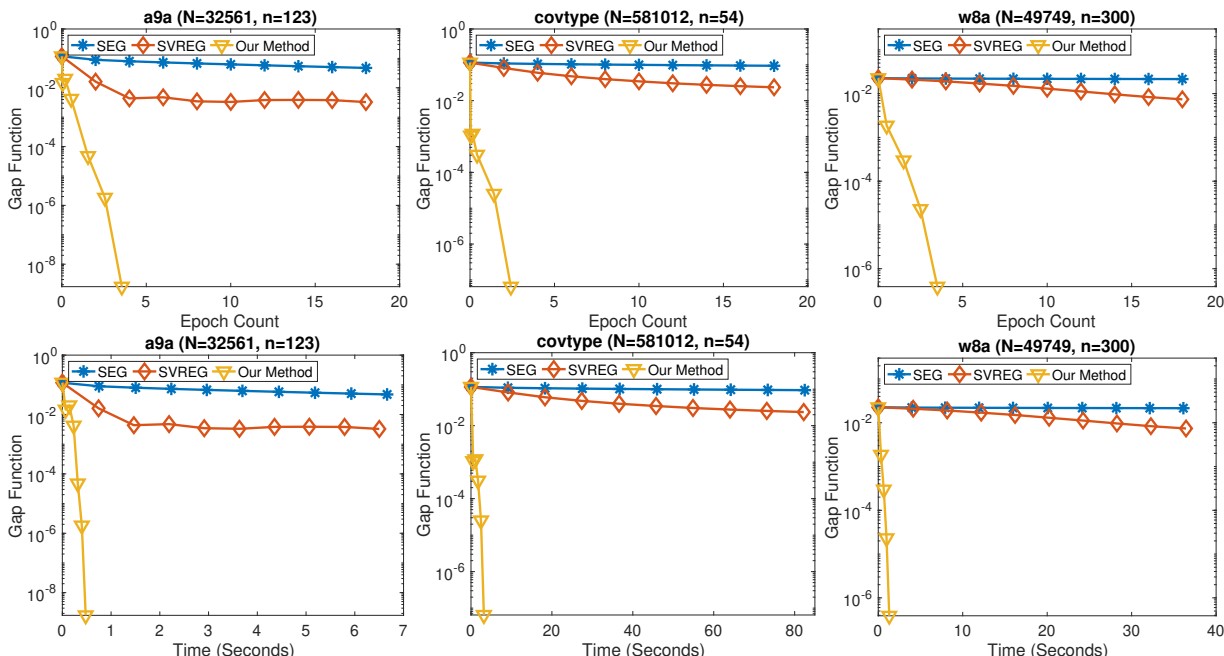

Figure 2: Performance of all algorithms with 3 LIBSVM datasets when $\rho = \frac{1}{N}$. The numerical results are presented in terms of iteration count (Top) and computational time (Bottom).

empirically as $\rho = \frac{1}{N}$ and we use 3 LIBSVM datasets[3] for AUC maximization. Since this min-max problem has a finite-sum structure, we can apply Algorithm 3 with uniform sampling. The baseline methods include stochastic first-order methods (Juditsky et al., 2011; Hsieh et al., 2019; Mertikopoulos et al., 2019) and stochastic variance-reduced first-order methods (Alacaoglu & Malitsky, 2022).[4] Among the variance-reduced methods proposed in Alacaoglu & Malitsky (2022), we include only one representative (which we refer to as SVREG) due to their comparable performance. We choose the stepsizes for stochastic first-order methods in the form of $\frac{c}{\sqrt{k+1}}$ where $c > 0$ is tuned using grid search and $k \geq 0$ is an iteration count. We choose the default parameters for variance-reduced first-order methods as in Alacaoglu & Malitsky (2022). For Algorithm 3, we choose $\kappa_m = 10^{-6}$, $\delta = 0.01$ and $|\mathcal{S}_k| = \frac{20 \log(d+3)}{\min\{\|F(\hat{\mathbf{z}}_k)\|^2, \|F(\mathbf{z}_k)\|^2\}}$. Unless stated otherwise, we use the same hyperparameters for all datasets. We use $\min\{\|F(\hat{\mathbf{z}}_k)\|^2, \|F(\mathbf{z}_k)\|^2\}$ instead of $\|F(\hat{\mathbf{z}}_k)\|^2$ for practical purpose and such choice does not violate our theoretical results. For the subproblem solving, we apply the safeguarded Newton-bisection/Newton variants described in Section 4.2.

Figure 2 shows that Algorithm 3 outperforms stochastic first-order methods in terms of solution quality: stochastic first-order methods barely move when Algorithm 3 makes significant progress. Notably, Algorithm 3 exhibits (super)-linear convergence as the iterates approach the optimal set regardless of subsampling and inexact subproblem solving. This intriguing property has been rigorously justified for subsampled Newton methods for convex optimization (Roosta-Khorasani & Mahoney, 2019), and extending such results to convex-concave min-max optimization is an interesting future direction.

**Hyperparameters and practical sensitivity.** We discuss the algorithmic hyperparameters and their practical sensitivity. In Algorithm 3, the main choices are: (i) the safeguard parameter $\kappa_m$ in Condition 4.2; (ii) the confidence parameter $\delta$ in the subsampling rule, which influences $|\mathcal{S}_k|$ only through logarithmic factors; and (iii) the stopping tolerance for the one-dimensional root-finding routine used to compute the step parameter in the update. Across all synthetic and real experiments, we use a single default configuration without dataset-specific tuning and observe stable behavior. Smaller $\kappa_m$ or a tighter root-finding tolerance

---

[3]https://www.csie.ntu.edu.tw/~cjlin/libsvm/
[4]The stochastic variants of EG and OGDA exhibit similar performance in iteration count and computational time so we include only one of them (which we refer to as SEG).

typically increases the number of inner linear solves but can improve the final accuracy, whereas looser settings reduce per-iteration cost while retaining stability due to the safeguarded Newton–bisection design. The parameter $\delta$ enters the sample-size schedule only via logarithmic terms, and setting $\delta$ to a small constant (e.g., $10^{-2}$) provides a reliable default in practice. The admissible inexactness level $\tau_k$ in Condition 4.1 depends on $(\kappa_m, \kappa_J, \rho)$ and $\|F(\hat{\mathbf{z}}_k)\|$, affecting constants and inner-loop work but not the overall structure: each outer iteration requires one Schur decomposition and only a few quasi-triangular linear solves, with doubly-logarithmic dependence on the target inner accuracy in the local Newton regime. While a systematic sensitivity sweep is an interesting direction for future work, these results suggest the method is robust and performs well with fixed defaults across the reported settings.

**Scope of empirical evaluation.**   Our experiments are designed to validate the practical behavior of our methods under both deterministic and finite-sum settings, with an emphasis on the regimes targeted by our theory, i.e., simultaneous inexactness in (i) constructing second-order information and (ii) solving the resulting subproblems. Beyond synthetic ones, we include AUC maximization on LIBSVM datasets as a non-quadratic finite-sum learning problem, and we evaluate the subsampled variant that constructs $J(\hat{\mathbf{z}}_k)$ from minibatches, thereby reflecting the inexact-Jacobian scenario studied in Section 4.3. In this finite-sum regime, the minibatch size and the schedule controlling the construction accuracy of $J(\hat{\mathbf{z}}_k)$ provide a principled and algorithmically explicit way to modulate the "Jacobian noise" induced by subsampling; correspondingly, our safeguarded globalization (acceptance and Newton–bisection fallback) is designed to remain stable under moderate inaccuracies while progressively tightening accuracy as higher solution precision is required.

We emphasize, however, that our experiments are not intended to be exhaustive across all large-scale ML tasks. A comprehensive benchmark suite would include more datasets and model classes, systematic sweeps over subsampling-induced noise levels, and comparisons to quasi-Newton and limited-memory alternatives, especially in large-scale regimes where factorization-based second-order methods are memory-limited and where baseline performance depends sensitively on choices such as curvature updates, damping/safeguarding, and stopping criteria. That being said, we phrase our empirical conclusions as evidence of stable behavior and competitive efficiency on the tested instances, rather than as a universal dominance claim; we view more benchmarking, quasi-Newton comparisons, and dedicated noise-sensitivity studies as future works.

## 6   Conclusion

We propose several inexact regularized Newton-type methods for finding a global saddle point of unconstrained convex-concave min-max optimization problems. Our inexact methods are guaranteed to achieve an iteration complexity of $O(\epsilon^{-2/3})$ and an improved complexity bound of $O((m + n)^{\omega}\epsilon^{-2/3} + (m + n)^2\epsilon^{-2/3}\log\log(1/\epsilon))$ where $\omega \approx 2.3728$ is the matrix multiplication constant. In addition, we show that our general framework and analysis yields the first class of subsampled Newton method for solving finite-sum min-max optimization problems with the same iteration complexity of $O(\epsilon^{-2/3})$. Future research directions include the extension of our methods to structured nonconvex-nonconcave min-max optimization problems and the customized implementation of our methods in real application problems.

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

# A  Related Work

Our work comes amid a surge of interest in optimization algorithms for a large class of emerging min-max optimization problems. For brevity, we focus on convex-concave settings and will leave other settings aside; see Section 2 in Lin et al. (2020) for a more thorough presentation of algorithms and complexity results.

Historically, a concrete instantiation of the convex-concave min-max optimization problem is the solution of $\min_{\mathbf{x} \in \Delta^m} \max_{\mathbf{y} \in \Delta^n} \mathbf{x}^\top A \mathbf{y}$ for $A \in \mathbb{R}^{m \times n}$ over the simplices $\Delta^m$ and $\Delta^n$. Spurred by von Neumann's theorem (Neumann, 1928), this problem provided the initial impetus for min-max optimization. Sion (1958) generalized von Neumann's result from bilinear cases to convex-concave cases and triggered a line of research on algorithms for convex-concave min-max optimization (Korpelevich, 1976; Nemirovski, 2004; Nesterov, 2007; Nedić & Ozdaglar, 2009; Mokhtari et al., 2020b). A notable result is that gradient descent ascent (GDA) with diminishing stepsizes can find an $\epsilon$-global saddle point within $O(\epsilon^{-2})$ iterations if the gradients are bounded over the feasible sets (Nedić & Ozdaglar, 2009).

Recent years have witnessed much progress on the design and analysis of first-order min-max optimization algorithms in bilinear cases and convex-concave cases. In the bilinear case, Daskalakis et al. (2018) proved the convergence of OGDA to a neighborhood of a global saddle point. Liang & Stokes (2019) used a dynamical system approach to establish the linear convergence of OGDA for the special case when the matrix $A$ is square and full rank. Mokhtari et al. (2020a) revisited a proximal point algorithmic framework and proposed an unified analysis for achieving the sharpest convergence rates of both EG and OGDA. In the convex-concave case, Nemirovski (2004) demonstrated that EG finds an $\epsilon$-global saddle point within $O(\epsilon^{-1})$ iterations when the feasible sets are convex and bounded. The same convergence guarantee was extended to unbounded sets (Monteiro & Svaiter, 2010; 2011) using the HPE method with different criteria. In a similar vein, Nesterov (2007) and Tseng (2008) proposed new algorithms and refined convergence analysis with the same convergence guarantee. Abernethy et al. (2021) presented a Hamiltonian gradient descent method with last-iterate convergence under a "sufficiently bilinear" condition. Focusing on a special class of min-max optimization problems with separable structure: $f(\mathbf{x}, \mathbf{y}) = g(\mathbf{x}) + \mathbf{x}^\top A \mathbf{y} - h(\mathbf{y})$, Chambolle & Pock (2011) introduced a primal-dual hybrid gradient method that converges to a global saddle point at the rate of $O(\epsilon^{-1})$ when the functions $g$ and $h$ are simple and convex. Nesterov (2005) introduced a smoothing technique and proved that the resulting algorithm achieved better dependence on some problem parameters for convex and bounded feasible sets. He & Monteiro (2016) and Kolossoski & Monteiro (2017) proved that such a result holds for unbounded sets and non-Euclidean metrics. In addition, Chen et al. (2014) proposed a new class of stochastic algorithms for solving separable min-max optimization problems when only noisy gradients are available and proved the optimal convergence guarantee.

There are also several papers on finite-sum min-max optimization and variance reduction (Balamurugan & Bach, 2016; Chavdarova et al., 2019; Carmon et al., 2019; Yang et al., 2020; Luo et al., 2021; Alacaoglu & Malitsky, 2022). In the strongly-convex-strongly-concave cases, Balamurugan & Bach (2016) studied stochastic variance-reduced variants of forward-backward methods and proved linear convergence. Chavdarova et al. (2019) combined EG with variance reduction and proved new convergence guarantee which was however less favorable than Balamurugan & Bach (2016). Recently, Luo et al. (2021) have considered unbalanced strongly-convex-strongly-concave cases and proposed variance-reduced first-order methods with an improved complexity bound. In the bilinear case, Carmon et al. (2019) proposed a randomized mirror-prox method with an improved complexity guarantee. However, extending their results to general convex-concave cases requires additional restrictive assumptions. Alacaoglu & Malitsky (2022) addressed this issue by leveraging a new variance reduction technique and their methods either match or improve the best-known complexities for solving convex-concave finite-sum min-max problems. Beyond the convex-concave cases, Yang et al. (2020) provided various variance-reduced first-order methods for solving structured nonconvex-concave min-max optimization problems.

Compared to first-order methods, there has been less research on second-order methods for min-max optimization problems with *global* convergence rate guarantee. In this context, we are aware of two research thrusts (Monteiro & Svaiter, 2012; Bullins & Lai, 2022; Huang et al., 2022; Adil et al., 2022; Lin & Jordan, 2023; 2025; Jiang & Mokhtari, 2025; Chen et al., 2025a). Our results contribute to this landscape by proposing the first explicit method that achieves the iteration complexity of $O(\epsilon^{-2/3})$ as well as a new

complexity bound of $O((m + n)^\omega \epsilon^{-2/3} + (m + n)^2 \epsilon^{-2/3} \log\log(1/\epsilon))$. As far as we know, the above complexity bound guarantees cannot be realized by other existing second-order min-max optimization methods with exact second-order information requirements. Subsequent to our work, new strategies were designed for improving second-order min-max optimization methods (Alves et al., 2024; Jiang et al., 2024; Chen et al., 2025b), thereby matching and even surpassing our results.

## B   Additional Proofs from Section 2

**Proof of Lemma 2.9.**   Note that (a) and (c) were proven in Nemirovski (2004), and it suffices to prove (b). By using the definition of $DF(\cdot)$ in Eq. (2.3), we have

$$(DF(\mathbf{z}) - DF(\mathbf{z}'))\mathbf{h} = \begin{bmatrix} I_m & \\ & -I_n \end{bmatrix} (\nabla^2 f(\mathbf{z}) - \nabla^2 f(\mathbf{z}'))\mathbf{h}. \tag{B.1}$$

This implies that $\|(DF(\mathbf{z}) - DF(\mathbf{z}'))\mathbf{h}\| = \|(\nabla^2 f(\mathbf{z}) - \nabla^2 f(\mathbf{z}'))\mathbf{h}\|$. Thus, we have

$$\|DF(\mathbf{z}) - DF(\mathbf{z}')\| = \sup_{\mathbf{h} \neq 0} \left\{ \tfrac{\|(DF(\mathbf{z}) - DF(\mathbf{z}'))\mathbf{h}\|}{\|\mathbf{h}\|} \right\} = \sup_{\mathbf{h} \neq 0} \left\{ \tfrac{\|(\nabla^2 f(\mathbf{z}) - \nabla^2 f(\mathbf{z}'))\mathbf{h}\|}{\|\mathbf{h}\|} \right\}.$$

This equality together with Assumption 2.5 implies the desired result in (b). $\qquad\square$

**Proof of Proposition 2.10.**   Using the definition of the operator $F(\cdot)$ in Eq. (2.2), we have

$$\sum_{i=1}^{k} \lambda_i (\mathbf{z}_i - \mathbf{z})^\top F(\mathbf{z}_i) = \sum_{i=1}^{k} \lambda_i ((\mathbf{x}_i - \mathbf{x})^\top \nabla_\mathbf{x} f(\mathbf{x}_i, \mathbf{y}_i) - (\mathbf{y}_i - \mathbf{y})^\top \nabla_\mathbf{y} f(\mathbf{x}_i, \mathbf{y}_i)).$$

Note that Assumption 2.4 implies that the function $f(\mathbf{x}, \mathbf{y})$ is a convex function of $\mathbf{x}$ for any $\mathbf{y} \in \mathbb{R}^n$ and a concave function of $\mathbf{y}$ for any $\mathbf{x} \in \mathbb{R}^m$. Then, we have

$$\begin{aligned} (\mathbf{x}_i - \mathbf{x})^\top \nabla_\mathbf{x} f(\mathbf{x}_i, \mathbf{y}_i) &\geq& f(\mathbf{x}_i, \mathbf{y}_i) - f(\mathbf{x}, \mathbf{y}_i), \\ (\mathbf{y}_i - \mathbf{y})^\top \nabla_\mathbf{y} f(\mathbf{x}_i, \mathbf{y}_i) &\leq& f(\mathbf{x}_i, \mathbf{y}_i) - f(\mathbf{x}_i, \mathbf{y}). \end{aligned}$$

Putting these pieces together with $\lambda_i > 0$ for all $1 \leq i \leq k$ yields that

$$\tfrac{1}{\sum_{i=1}^{k} \lambda_i} \left( \sum_{i=1}^{k} \lambda_i (\mathbf{z}_i - \mathbf{z})^\top F(\mathbf{z}_i) \right) \geq \tfrac{1}{\sum_{i=1}^{k} \lambda_i} \left( \sum_{i=1}^{k} \lambda_i (f(\mathbf{x}_i, \mathbf{y}) - f(\mathbf{x}, \mathbf{y}_i)) \right). \tag{B.2}$$

Using the definition of $(\bar{\mathbf{x}}_k, \bar{\mathbf{y}}_k)$ in Eq. (2.1) and that $f$ is convex-concave, we have

$$\tfrac{1}{\sum_{i=1}^{k} \lambda_i} \left( \sum_{i=1}^{k} \lambda_i f(\mathbf{x}_i, \mathbf{y}) \right) \geq f(\bar{\mathbf{x}}_k, \mathbf{y}), \qquad \tfrac{1}{\sum_{i=1}^{k} \lambda_i} \left( \sum_{i=1}^{k} \lambda_i f(\mathbf{x}, \mathbf{y}_i) \right) \leq f(\mathbf{x}, \bar{\mathbf{y}}_k).$$

Plugging these two inequalities in Eq. (B.2) yields the desired inequality. $\qquad\square$

## C   Additional Proofs from Section 3

The key step is to define a Lyapunov function for Algorithm 1: $\mathcal{E}_t = \frac{1}{2}\|\hat{\mathbf{z}}_t - \mathbf{z}_0\|^2$, which can be used to prove technical results that pertain to the convergence property of Algorithm 1. The first lemma gives us a descent inequality.

**Lemma C.1** *Suppose that Assumptions 2.4 and 2.5 hold. Then*

$$\sum_{k=1}^{t} \lambda_k (\mathbf{z}_k - \mathbf{z})^\top F(\mathbf{z}_k) \leq \mathcal{E}_0 - \mathcal{E}_t + (\mathbf{z}_0 - \hat{\mathbf{z}}_t)^\top (\mathbf{z}_0 - \mathbf{z}) - \tfrac{1}{24} \left( \sum_{k=1}^{t} \|\mathbf{z}_k - \hat{\mathbf{z}}_{k-1}\|^2 \right).$$

*Proof.* Using the definition of the Lyapunov function, we have

$$\mathcal{E}_k - \mathcal{E}_{k-1} = \tfrac{1}{2}\|\hat{\mathbf{z}}_k - \mathbf{z}_0\|^2 - \tfrac{1}{2}\|\hat{\mathbf{z}}_{k-1} - \mathbf{z}_0\|^2 = (\hat{\mathbf{z}}_k - \hat{\mathbf{z}}_{k-1})^\top(\hat{\mathbf{z}}_k - \mathbf{z}_0) - \tfrac{1}{2}\|\hat{\mathbf{z}}_k - \hat{\mathbf{z}}_{k-1}\|^2. \quad (\text{C.1})$$

Plugging $\hat{\mathbf{z}}_k = \hat{\mathbf{z}}_{k-1} - \lambda_k F(\mathbf{z}_k)$ into Eq. (C.1) yields

$$\begin{aligned}
\mathcal{E}_k - \mathcal{E}_{k-1} &\leq \lambda_k(\mathbf{z}_0 - \hat{\mathbf{z}}_k)^\top F(\mathbf{z}_k) - \tfrac{1}{2}\|\hat{\mathbf{z}}_k - \hat{\mathbf{z}}_{k-1}\|^2 \\
&= \lambda_k(\mathbf{z}_0 - \mathbf{z})^\top F(\mathbf{z}_k) + \lambda_k(\mathbf{z} - \mathbf{z}_k)^\top F(\mathbf{z}_k) + \lambda_k(\mathbf{z}_k - \hat{\mathbf{z}}_k)^\top F(\mathbf{z}_k) - \tfrac{1}{2}\|\hat{\mathbf{z}}_k - \hat{\mathbf{z}}_{k-1}\|^2.
\end{aligned}$$

Summing up the above inequality over $k = 1, 2, \ldots, t$ yields

$$\sum_{k=1}^{t} \lambda_k(\mathbf{z}_k - \mathbf{z})^\top F(\mathbf{z}_k) \leq \mathcal{E}_0 - \mathcal{E}_t + \underbrace{\sum_{k=1}^{t} \lambda_k(\mathbf{z}_0 - \mathbf{z})^\top F(\mathbf{z}_k)}_{\mathbf{I}} + \underbrace{\sum_{k=1}^{t} \lambda_k(\mathbf{z}_k - \hat{\mathbf{z}}_k)^\top F(\mathbf{z}_k) - \tfrac{1}{2}\|\hat{\mathbf{z}}_k - \hat{\mathbf{z}}_{k-1}\|^2}_{\mathbf{II}}. \quad (\text{C.2})$$

By using the relationship $\hat{\mathbf{z}}_k = \hat{\mathbf{z}}_{k-1} - \lambda_k F(\mathbf{z}_k)$ again, we have

$$\mathbf{I} = \sum_{k=1}^{t} \lambda_k(\mathbf{z}_0 - \mathbf{z})^\top F(\mathbf{z}_k) = \sum_{k=1}^{t} (\hat{\mathbf{z}}_{k-1} - \hat{\mathbf{z}}_k)^\top(\mathbf{z}_0 - \mathbf{z}) = (\hat{\mathbf{z}}_0 - \hat{\mathbf{z}}_t)^\top(\mathbf{z}_0 - \mathbf{z}). \quad (\text{C.3})$$

In Algorithm 1, we compute $\Delta\mathbf{z}_k$ as an exact solution of the nonlinear equation problem given by

$$F(\hat{\mathbf{z}}_k) + DF(\hat{\mathbf{z}}_k)\Delta\mathbf{z}_k + 6\rho\|\Delta\mathbf{z}_k\|\Delta\mathbf{z}_k = \mathbf{0}. \quad (\text{C.4})$$

Using $\mathbf{z}_k = \hat{\mathbf{z}}_{k-1} + \Delta\mathbf{z}_{k-1}$ and Lemma 2.9, we have

$$\|F(\mathbf{z}_k) - F(\hat{\mathbf{z}}_{k-1}) - DF(\hat{\mathbf{z}}_{k-1})\Delta\mathbf{z}_{k-1}\| \leq \tfrac{\rho}{2}\|\Delta\mathbf{z}_{k-1}\|^2 \quad (\text{C.5})$$

so it suffices to decompose $(\mathbf{z}_k - \hat{\mathbf{z}}_k)^\top F(\mathbf{z}_k)$ and bound this term using Eq. (C.4) and (C.5). Indeed, to that end, we have

$$\begin{aligned}
(\mathbf{z}_k - \hat{\mathbf{z}}_k)^\top F(\mathbf{z}_k) &\leq \tfrac{\rho}{2}\|\Delta\mathbf{z}_{k-1}\|^2\|\mathbf{z}_k - \hat{\mathbf{z}}_k\| - 6\rho\|\Delta\mathbf{z}_{k-1}\|(\mathbf{z}_k - \hat{\mathbf{z}}_k)^\top\Delta\mathbf{z}_{k-1} \\
&\leq \tfrac{\rho}{2}(\|\Delta\mathbf{z}_{k-1}\|^3 + \|\Delta\mathbf{z}_{k-1}\|^2\|\hat{\mathbf{z}}_{k-1} - \hat{\mathbf{z}}_k\|) - 6\rho\|\Delta\mathbf{z}_{k-1}\|(\mathbf{z}_k - \hat{\mathbf{z}}_k)^\top\Delta\mathbf{z}_{k-1}.
\end{aligned}$$

Note that we have $(\mathbf{z}_k - \hat{\mathbf{z}}_k)^\top\Delta\mathbf{z}_{k-1} \geq \|\Delta\mathbf{z}_{k-1}\|^2 - \|\Delta\mathbf{z}_{k-1}\|\|\hat{\mathbf{z}}_{k-1} - \hat{\mathbf{z}}_k\|$. This implies

$$\|\Delta\mathbf{z}_{k-1}\|(\mathbf{z}_k - \hat{\mathbf{z}}_k)^\top\Delta\mathbf{z}_{k-1} \geq \|\Delta\mathbf{z}_{k-1}\|^3 - \|\Delta\mathbf{z}_{k-1}\|^2\|\hat{\mathbf{z}}_{k-1} - \hat{\mathbf{z}}_k\|.$$

Putting these pieces together yields

$$(\mathbf{z}_k - \hat{\mathbf{z}}_k)^\top F(\mathbf{z}_k) \leq \tfrac{13\rho}{2}\|\Delta\mathbf{z}_{k-1}\|^2\|\hat{\mathbf{z}}_{k-1} - \hat{\mathbf{z}}_k\| - \tfrac{11\rho}{2}\|\Delta\mathbf{z}_{k-1}\|^3.$$

Since $\tfrac{1}{33} \leq \lambda_k\rho\|\Delta\mathbf{z}_{k-1}\| \leq \tfrac{1}{13}$ for all $k \geq 1$, we have

$$\begin{aligned}
\mathbf{II} &\leq \sum_{k=1}^{t}\left(\tfrac{13\lambda_k\rho}{2}\|\Delta\mathbf{z}_{k-1}\|^2\|\hat{\mathbf{z}}_{k-1} - \hat{\mathbf{z}}_k\| - \tfrac{1}{2}\|\hat{\mathbf{z}}_{k-1} - \hat{\mathbf{z}}_k\|^2 - \tfrac{11\lambda_k\rho}{2}\|\Delta\mathbf{z}_{k-1}\|^3\right) \\
&\leq \sum_{k=1}^{t}\left(\tfrac{1}{2}\|\Delta\mathbf{z}_{k-1}\|\|\hat{\mathbf{z}}_{k-1} - \hat{\mathbf{z}}_k\| - \tfrac{1}{2}\|\hat{\mathbf{z}}_{k-1} - \hat{\mathbf{z}}_k\|^2 - \tfrac{1}{6}\|\Delta\mathbf{z}_{k-1}\|^2\right) \quad (\text{C.6}) \\
&\leq \sum_{k=1}^{t}\left(\max_{\eta\geq 0}\left\{\tfrac{1}{2}\|\Delta\mathbf{z}_{k-1}\|\eta - \tfrac{1}{2}\eta^2\right\} - \tfrac{1}{6}\|\Delta\mathbf{z}_{k-1}\|^2\right) = -\tfrac{1}{24}\left(\sum_{k=1}^{t}\|\Delta\mathbf{z}_{k-1}\|^2\right).
\end{aligned}$$

Plugging Eq. (C.3) and (C.6) into Eq. (C.2) and using $\hat{\mathbf{z}}_0 = \mathbf{z}_0$ and $\Delta\mathbf{z}_{k-1} = \mathbf{z}_k - \hat{\mathbf{z}}_{k-1}$ yields

$$\sum_{k=1}^{t} \lambda_k(\mathbf{z}_k - \mathbf{z})^\top F(\mathbf{z}_k) \leq \mathcal{E}_0 - \mathcal{E}_t + (\mathbf{z}_0 - \hat{\mathbf{z}}_t)^\top(\mathbf{z}_0 - \mathbf{z}) - \tfrac{1}{24}\left(\sum_{k=1}^{t}\|\mathbf{z}_k - \hat{\mathbf{z}}_{k-1}\|^2\right).$$

This completes the proof. $\square$

**Lemma C.2** *Suppose that Assumptions 2.4 and 2.5 hold. Then, we have $\|\hat{\mathbf{z}}_t - \mathbf{z}_0\| \leq 2\|\mathbf{z}_0 - \mathbf{z}^\star\|$ and*

$$\sum_{k=1}^{t} \lambda_k (\mathbf{z}_k - \mathbf{z})^\top F(\mathbf{z}_k) \leq \tfrac{1}{2}\|\mathbf{z}_0 - \mathbf{z}\|^2, \quad \sum_{k=1}^{t} \|\mathbf{z}_k - \hat{\mathbf{z}}_{k-1}\|^2 \leq 12\|\mathbf{z}_0 - \mathbf{z}^\star\|^2,$$

*where $\mathbf{z} \in \mathbb{R}^{m+n}$ is any point and $\mathbf{z}^\star$ is a global saddle point.*

*Proof.* Since $\hat{\mathbf{z}}_0 = \mathbf{z}_0$, we have

$$\mathcal{E}_0 - \mathcal{E}_t + (\mathbf{z}_0 - \hat{\mathbf{z}}_t)^\top (\mathbf{z}_0 - \mathbf{z}) \leq -\tfrac{1}{2}\|\hat{\mathbf{z}}_t - \mathbf{z}_0\|^2 + \tfrac{1}{2}\|\hat{\mathbf{z}}_t - \mathbf{z}_0\|^2 + \tfrac{1}{2}\|\mathbf{z}_0 - \mathbf{z}\|^2 = \tfrac{1}{2}\|\mathbf{z}_0 - \mathbf{z}\|^2.$$

This together with Lemma C.1 yields

$$\sum_{k=1}^{t} \lambda_k (\mathbf{z}_k - \mathbf{z})^\top F(\mathbf{z}_k) \leq \tfrac{1}{2}\|\mathbf{z}_0 - \mathbf{z}\|^2 - \tfrac{1}{24}\left(\sum_{k=1}^{t} \|\mathbf{z}_k - \hat{\mathbf{z}}_{k-1}\|^2\right) \leq \tfrac{1}{2}\|\mathbf{z}_0 - \mathbf{z}\|^2.$$

Since $\mathbf{z}^\star$ is a global saddle point, we have $(\mathbf{z}_k - \mathbf{z}^\star)^\top F(\mathbf{z}_k) \geq 0$ for all $k \geq 1$. Then, we have

$$\sum_{k=1}^{t} \|\mathbf{z}_k - \hat{\mathbf{z}}_{k-1}\|^2 \leq 12\|\mathbf{z}_0 - \mathbf{z}^\star\|^2.$$

Further, Lemma C.1 with $\mathbf{z} = \mathbf{z}^\star$ implies

$$\mathcal{E}_0 - \mathcal{E}_t + (\mathbf{z}_0 - \hat{\mathbf{z}}_t)^\top (\mathbf{z}_0 - \mathbf{z}^\star) \geq \sum_{k=1}^{t} \lambda_k (\mathbf{z}_k - \mathbf{z}^\star)^\top F(\mathbf{z}_k) + \tfrac{1}{24}\left(\sum_{k=1}^{t} \|\mathbf{z}_k - \hat{\mathbf{z}}_{k-1}\|^2\right) \geq 0.$$

Using Young's inequality, we have

$$0 \leq -\tfrac{1}{2}\|\hat{\mathbf{z}}_t - \mathbf{z}_0\|^2 + \tfrac{1}{4}\|\hat{\mathbf{z}}_t - \mathbf{z}_0\|^2 + \|\mathbf{z}_0 - \mathbf{z}^\star\|^2 = -\tfrac{1}{4}\|\hat{\mathbf{z}}_t - \mathbf{z}_0\|^2 + \|\mathbf{z}_0 - \mathbf{z}^\star\|^2.$$

This completes the proof. $\qquad\square$

**Lemma C.3** *Suppose that Assumptions 2.4 and 2.5 hold. Then, for every integer $T \geq 1$, we have*

$$\sum_{k=1}^{T} \lambda_k \geq \frac{T^{\frac{3}{2}}}{66\sqrt{3}\rho\|\mathbf{z}_0 - \mathbf{z}^\star\|},$$

*where $\mathbf{z}^\star$ is a global saddle point.*

*Proof.* Without loss of generality, we assume that $\mathbf{z}_0 \neq \mathbf{z}^\star$. Then, Lemma C.2 implies

$$\sum_{k=1}^{t} (\lambda_k)^{-2}(\tfrac{1}{33\rho})^2 \leq \sum_{k=1}^{t} (\lambda_k)^{-2}(\lambda_k\|\mathbf{z}_k - \hat{\mathbf{z}}_{k-1}\|)^2 = \sum_{k=1}^{t} \|\mathbf{z}_k - \hat{\mathbf{z}}_{k-1}\|^2 \leq 12\|\mathbf{z}_0 - \mathbf{z}^\star\|^2.$$

By the Hölder inequality, we have

$$\sum_{k=1}^{t} 1 = \sum_{k=1}^{t} \left((\lambda_k)^{-2}\right)^{\frac{1}{3}} (\lambda_k)^{\frac{2}{3}} \leq \left(\sum_{k=1}^{t}(\lambda_k)^{-2}\right)^{\frac{1}{3}} \left(\sum_{k=1}^{t}\lambda_k\right)^{\frac{2}{3}}.$$

Putting these pieces together yields

$$t \leq \left(66\sqrt{3}\rho\|\mathbf{z}_0 - \mathbf{z}^\star\|\right)^{\frac{2}{3}} \left(\sum_{k=1}^{t}\lambda_k\right)^{\frac{2}{3}}.$$

Letting $t = T$ and rearranging yields the desired result. $\qquad\square$

**Proof of Theorem 3.1** By Lemma C.2, we have

$$\|\mathbf{z}_{k+1} - \hat{\mathbf{z}}_k\|^2 \leq 12\|\mathbf{z}_0 - \mathbf{z}^\star\|^2, \quad \|\hat{\mathbf{z}}_k - \mathbf{z}_0\| \leq 2\|\mathbf{z}_0 - \mathbf{z}^\star\|, \quad \text{for all } k \geq 0.$$

This implies that $\|\mathbf{z}_k - \mathbf{z}_0\| \leq 6\|\mathbf{z}_0 - \mathbf{z}^\star\|$ for all $k \geq 0$. Putting these pieces together yields that both $\{\mathbf{z}_k\}_{k \geq 0}$ and $\{\hat{\mathbf{z}}_k\}_{k \geq 0}$ are bounded by a constant; indeed, we have $\|\hat{\mathbf{z}}_k - \mathbf{z}^\star\| \leq 3\|\mathbf{z}_0 - \mathbf{z}^\star\| \leq \beta$ and $\|\mathbf{z}_k - \mathbf{z}^\star\| \leq 7\|\mathbf{z}_0 - \mathbf{z}^\star\| = \beta$. For every integer $T \geq 1$, Lemma C.2 also implies

$$\sum_{k=1}^{T} \lambda_k (\mathbf{z}_k - \mathbf{z})^\top F(\mathbf{z}_k) \leq \tfrac{1}{2}\|\mathbf{z}_0 - \mathbf{z}\|^2.$$

By Proposition 2.10, we have

$$f(\bar{\mathbf{x}}_T, \mathbf{y}) - f(\mathbf{x}, \bar{\mathbf{y}}_T) \leq \frac{1}{\sum_{k=1}^{T} \lambda_k} \left( \sum_{k=1}^{T} \lambda_k (\mathbf{z}_k - \mathbf{z})^\top F(\mathbf{z}_k) \right).$$

Putting these pieces together yields

$$f(\bar{\mathbf{x}}_T, \mathbf{y}) - f(\mathbf{x}, \bar{\mathbf{y}}_T) \leq \frac{1}{2(\sum_{k=1}^{T} \lambda_k)} \|\mathbf{z}_0 - \mathbf{z}\|^2.$$

This together with Lemma C.3 yields

$$f(\bar{\mathbf{x}}_T, \mathbf{y}) - f(\mathbf{x}, \bar{\mathbf{y}}_T) \leq \frac{33\sqrt{3}\rho\|\mathbf{z}_0 - \mathbf{z}^\star\|\|\mathbf{z}_0 - \mathbf{z}\|^2}{T^{3/2}}.$$

Since $\|\mathbf{z}_k - \mathbf{z}^\star\| \leq \beta$ for all $k \geq 0$, we have $\|\bar{\mathbf{z}}_T - \mathbf{z}^\star\| \leq \beta$. By the definition of the restricted gap function, we have

$$\text{gap}(\bar{\mathbf{z}}_T, \beta) \leq \frac{33\sqrt{3}\rho\|\mathbf{z}_0 - \mathbf{z}^\star\|(\|\mathbf{z}_0 - \mathbf{z}^\star\| + \beta)^2}{T^{3/2}} = \frac{2112\sqrt{3}\rho\|\mathbf{z}_0 - \mathbf{z}^\star\|^3}{T^{3/2}}.$$

Therefore, we conclude from the above inequality that there exists some $T > 0$ such that the output $\hat{\mathbf{z}} = \text{Newton-MinMax}(\mathbf{z}_0, \rho, T)$ satisfies that $\text{gap}(\hat{\mathbf{z}}, \beta) \leq \epsilon$ and the total number of iterations is bounded by $O(\rho^{2/3}\|\mathbf{z}_0 - \mathbf{z}^\star\|^2 \epsilon^{-2/3})$. □

# D    Additional Proofs from Section 4

In the subsequent analysis, we use the Lyapunov function: $\mathcal{E}_t = \frac{1}{2}\|\hat{\mathbf{z}}_t - \mathbf{z}_0\|^2$. The first lemma gives a descent inequality which is analogous to that in Lemma C.1.

**Lemma D.1** *Suppose that Assumption 2.4 and 2.5 hold and*

$$0 < \tau_k \leq \min\{\tau_0, \tfrac{\rho(1 - \kappa_m)}{4(\kappa_J + 6\rho)}\|F(\hat{\mathbf{z}}_k)\|\}, \quad \text{for all } k \geq 0.$$

*Then, we have*

$$\sum_{k=1}^{t} \lambda_k (\mathbf{z}_k - \mathbf{z})^\top F(\mathbf{z}_k) \leq \mathcal{E}_0 - \mathcal{E}_t + (\mathbf{z}_0 - \hat{\mathbf{z}}_t)^\top (\mathbf{z}_0 - \mathbf{z}) - \tfrac{1}{24} \left( \sum_{k=1}^{t} \|\mathbf{z}_k - \hat{\mathbf{z}}_{k-1}\|^2 \right).$$

*Proof.* By using the same argument as used in Lemma C.1, we have

$$\sum_{k=1}^{t} \lambda_k (\mathbf{z}_k - \mathbf{z})^\top F(\mathbf{z}_k) \leq \mathcal{E}_0 - \mathcal{E}_t + (\hat{\mathbf{z}}_0 - \hat{\mathbf{z}}_t)^\top (\mathbf{z}_0 - \mathbf{z}) + \sum_{k=1}^{t} \lambda_k (\mathbf{z}_k - \hat{\mathbf{z}}_k)^\top F(\mathbf{z}_k) - \tfrac{1}{2}\|\hat{\mathbf{z}}_k - \hat{\mathbf{z}}_{k-1}\|^2. \quad \text{(D.1)}$$

In what follows, we bound $\sum_{k=1}^{t} \lambda_k (\mathbf{z}_k - \hat{\mathbf{z}}_k)^\top F(\mathbf{z}_k) - \frac{1}{2}\|\hat{\mathbf{z}}_k - \hat{\mathbf{z}}_{k-1}\|^2$. In Algorithm 2, we compute $\Delta\mathbf{z}_k$ such that it is an *inexact* solution of the nonlinear equation problem given by Eq. (4.1) under Conditions 4.1 and 4.2. Note that we have

$$\|F(\hat{\mathbf{z}}_k) + J(\hat{\mathbf{z}}_k)\Delta\mathbf{z}_k + 6\rho\|\Delta\mathbf{z}_k\|\Delta\mathbf{z}_k\| \leq \kappa_m \cdot \min\{\|\Delta\mathbf{z}_k\|^2, \|F(\hat{\mathbf{z}}_k)\|\}. \quad \text{(D.2)}$$

Using $\mathbf{z}_k = \hat{\mathbf{z}}_{k-1} + \Delta\mathbf{z}_{k-1}$ and Lemma 2.9, we have

$$\|F(\mathbf{z}_k) - F(\hat{\mathbf{z}}_{k-1}) - DF(\hat{\mathbf{z}}_{k-1})\Delta\mathbf{z}_{k-1}\| \le \tfrac{\rho}{2}\|\Delta\mathbf{z}_{k-1}\|^2. \tag{D.3}$$

It suffices to decompose $(\mathbf{z}_k - \hat{\mathbf{z}}_k)^\top F(\mathbf{z}_k)$ and bound this term using Condition 4.1, Eq. (D.2) and (D.3). Indeed, we have

$$(\mathbf{z}_k - \hat{\mathbf{z}}_k)^\top F(\mathbf{z}_k) \le \|\mathbf{z}_k - \hat{\mathbf{z}}_k\|\|F(\mathbf{z}_k) - F(\hat{\mathbf{z}}_{k-1}) - DF(\hat{\mathbf{z}}_{k-1})\Delta\mathbf{z}_{k-1}\|$$
$$+\|\mathbf{z}_k - \hat{\mathbf{z}}_k\|\|F(\hat{\mathbf{z}}_{k-1}) + J(\hat{\mathbf{z}}_{k-1})\Delta\mathbf{z}_{k-1} + 6\rho\|\Delta\mathbf{z}_{k-1}\|\Delta\mathbf{z}_{k-1}\|$$
$$+\|\mathbf{z}_k - \hat{\mathbf{z}}_k\|\|(DF(\hat{\mathbf{z}}_{k-1}) - J(\hat{\mathbf{z}}_{k-1}))\Delta\mathbf{z}_{k-1}\| - 6\rho\|\Delta\mathbf{z}_{k-1}\|(\mathbf{z}_k - \hat{\mathbf{z}}_k)^\top\Delta\mathbf{z}_{k-1}.$$

The first and second terms are bounded using Eq. (D.2) and (D.3). For the third term, we derive from Condition 4.1 that $\|(DF(\hat{\mathbf{z}}_{k-1}) - J(\hat{\mathbf{z}}_{k-1}))\Delta\mathbf{z}_{k-1}\| \le \tau_{k-1}\|\Delta\mathbf{z}_{k-1}\|$. The fourth term is bounded using the same argument from the proof of Lemma C.1. Putting these pieces together yields that

$$(\mathbf{z}_k - \hat{\mathbf{z}}_k)^\top F(\mathbf{z}_k) \le \|\mathbf{z}_k - \hat{\mathbf{z}}_k\|\left((\tfrac{\rho}{2} + \kappa_m)\|\Delta\mathbf{z}_{k-1}\|^2 + \tau_{k-1}\|\Delta\mathbf{z}_{k-1}\|\right) - 6\rho\|\Delta\mathbf{z}_{k-1}\|^3 + 6\rho\|\Delta\mathbf{z}_{k-1}\|^2\|\hat{\mathbf{z}}_{k-1} - \hat{\mathbf{z}}_k\|. \tag{D.4}$$

We claim that

$$\left(\tfrac{\rho}{2} + \kappa_m\right)\|\Delta\mathbf{z}_{k-1}\|^2 + \tau_{k-1}\|\Delta\mathbf{z}_{k-1}\| \le \rho\|\Delta\mathbf{z}_{k-1}\|^2. \tag{D.5}$$

Indeed, for the case of $\|\Delta\mathbf{z}_{k-1}\| \ge 1$, we have

$$\left(\tfrac{\rho}{2} + \kappa_m\right)\|\Delta\mathbf{z}_{k-1}\|^2 + \tau_{k-1}\|\Delta\mathbf{z}_{k-1}\| \le \left(\tfrac{\rho}{2} + \kappa_m + \tau_{k-1}\right)\|\Delta\mathbf{z}_{k-1}\|^2.$$

which together with the fact that $0 < \kappa_m < \min\{1, \tfrac{\rho}{4}\}$ and $\tau_{k-1} \le \tau_0 < \tfrac{\rho}{4}$ can yield Eq. (D.5). Otherwise, we have $\|\Delta\mathbf{z}_{k-1}\| < 1$ and obtain from Conditions 4.1 and 4.2 that

$$\kappa_m\|F(\hat{\mathbf{z}}_{k-1})\| \ge \|F(\hat{\mathbf{z}}_{k-1}) + J(\hat{\mathbf{z}}_{k-1})\Delta\mathbf{z}_{k-1} + 6\rho\|\Delta\mathbf{z}_{k-1}\|\Delta\mathbf{z}_{k-1}\|$$
$$\ge \|F(\hat{\mathbf{z}}_{k-1})\| - \kappa_J\|\Delta\mathbf{z}_{k-1}\| - 6\rho\|\Delta\mathbf{z}_{k-1}\|^2$$
$$\ge \|F(\hat{\mathbf{z}}_{k-1})\| - (\kappa_J + 6\rho)\|\Delta\mathbf{z}_{k-1}\|.$$

Rearranging the above inequality and using $0 \le \tau_{k-1} \le \frac{\rho(1-\kappa_m)}{4(\kappa_J+6\rho)}\|F(\hat{\mathbf{z}}_{k-1})\|$ yields

$$\|\Delta\mathbf{z}_{k-1}\| \ge \tfrac{1-\kappa_m}{\kappa_J+6\rho}\|F(\hat{\mathbf{z}}_{k-1})\| \ge \tfrac{4\tau_{k-1}}{\rho}.$$

Since $0 < \kappa_m < \min\{1, \tfrac{\rho}{4}\}$ again, we get Eq. (D.5) as follows,

$$\left(\tfrac{\rho}{2} + \kappa_m\right)\|\Delta\mathbf{z}_{k-1}\|^2 + \tau_{k-1}\|\Delta\mathbf{z}_{k-1}\| \le \left(\tfrac{\rho}{2} + \kappa_m + \tfrac{\tau_{k-1}}{\|\Delta\mathbf{z}_{k-1}\|}\right)\|\Delta\mathbf{z}_{k-1}\|^2 \le \rho\|\Delta\mathbf{z}_{k-1}\|^2.$$

Plugging Eq. (D.5) into Eq. (D.4) and using $\|\mathbf{z}_k - \hat{\mathbf{z}}_k\| \le \|\Delta\mathbf{z}_{k-1}\| + \|\hat{\mathbf{z}}_{k-1} - \hat{\mathbf{z}}_k\|$ yields

$$(\mathbf{z}_k - \hat{\mathbf{z}}_k)^\top F(\mathbf{z}_k) \le 7\rho\|\Delta\mathbf{z}_{k-1}\|^2\|\|\hat{\mathbf{z}}_{k-1} - \hat{\mathbf{z}}_k\| - 5\rho\|\Delta\mathbf{z}_{k-1}\|^3.$$

Since $\tfrac{1}{30} \le \lambda_k\rho\|\Delta\mathbf{z}_{k-1}\| \le \tfrac{1}{14}$ for all $k \ge 1$, we have

$$\sum_{k=1}^{t}\lambda_k(\mathbf{z}_k - \hat{\mathbf{z}}_k)^\top F(\mathbf{z}_k) - \tfrac{1}{2}\|\hat{\mathbf{z}}_k - \hat{\mathbf{z}}_{k-1}\|^2$$

$$\le \sum_{k=1}^{t}\left(7\lambda_k\rho\|\Delta\mathbf{z}_{k-1}\|^2\|\|\hat{\mathbf{z}}_{k-1} - \hat{\mathbf{z}}_k\| - \tfrac{1}{2}\|\hat{\mathbf{z}}_{k-1} - \hat{\mathbf{z}}_k\|^2 - 5\lambda_k\rho\|\Delta\mathbf{z}_{k-1}\|^3\right)$$

$$\le \sum_{k=1}^{t}\left(\tfrac{1}{2}\|\Delta\mathbf{z}_{k-1}\|\|\hat{\mathbf{z}}_{k-1} - \hat{\mathbf{z}}_k\| - \tfrac{1}{2}\|\hat{\mathbf{z}}_{k-1} - \hat{\mathbf{z}}_k\|^2 - \tfrac{1}{6}\|\Delta\mathbf{z}_{k-1}\|^2\right)$$

$$\le \sum_{k=1}^{t}\left(\max_{\eta \ge 0}\left\{\tfrac{1}{2}\|\Delta\mathbf{z}_{k-1}\|\eta - \tfrac{1}{2}\eta^2\right\} - \tfrac{1}{6}\|\Delta\mathbf{z}_{k-1}\|^2\right) = -\tfrac{1}{24}\left(\sum_{k=1}^{t}\|\Delta\mathbf{z}_{k-1}\|^2\right).$$

Therefore, we conclude from Eq. (D.1), $\hat{\mathbf{z}}_0 = \mathbf{z}_0$ and $\Delta\mathbf{z}_{k-1} = \mathbf{z}_k - \hat{\mathbf{z}}_{k-1}$ that

$$\sum_{k=1}^{t}\lambda_k(\mathbf{z}_k - \mathbf{z})^\top F(\mathbf{z}_k) \le \mathcal{E}_0 - \mathcal{E}_t + (\mathbf{z}_0 - \hat{\mathbf{z}}_t)^\top(\mathbf{z}_0 - \mathbf{z}) - \tfrac{1}{24}\left(\sum_{k=1}^{t}\|\mathbf{z}_k - \hat{\mathbf{z}}_{k-1}\|^2\right).$$

This completes the proof. $\qquad\square$

**Proof of Theorem 4.1.** Since the descent inequalities in Lemmas C.1 and D.1 are the same, Lemma C.2 and C.3 also hold true for Algorithm 2. As such, we can apply the same argument used for proving Theorem 3.1 and have

$$\|\hat{\mathbf{z}}_k - \mathbf{z}^\star\| \leq 3\|\mathbf{z}_0 - \mathbf{z}^\star\| \leq \beta, \quad \|\mathbf{z}_k - \mathbf{z}^\star\| \leq 7\|\mathbf{z}_0 - \mathbf{z}^\star\| = \beta,$$

and

$$\mathrm{gap}(\bar{\mathbf{z}}_T, \beta) \leq \tfrac{2112\sqrt{3}\rho\|\mathbf{z}_0 - \mathbf{z}^\star\|^3}{T^{3/2}}.$$

Therefore, we conclude from the above inequality that there exists some $T > 0$ such that the output $\hat{\mathbf{z}} = $ Inexact-Newton-MinMax$(\mathbf{z}_0, \rho, T)$ satisfies that $\mathrm{gap}(\hat{\mathbf{z}}, \beta) \leq \epsilon$ and the total number of iterations is bounded by $O(\rho^{2/3}\|\mathbf{z}_0 - \mathbf{z}^\star\|^2\epsilon^{-2/3})$. $\qquad\square$

**Proof of Proposition 4.4.** Since $\psi(\lambda) = \|\Delta\mathbf{z}(\lambda)\|^2$, we have

$$\psi'(\lambda) = 2\Delta\mathbf{z}(\lambda)^\top \nabla_\lambda \Delta\mathbf{z}(\lambda), \quad \psi''(\lambda) = 2\|\nabla_\lambda \Delta\mathbf{z}(\lambda)\|^2 + 2\Delta\mathbf{z}(\lambda)^\top \nabla_{\lambda\lambda}^2 \Delta\mathbf{z}(\lambda).$$

By differentiating the equation $(J(\hat{\mathbf{z}}) + \lambda I)\Delta\mathbf{z}(\lambda) = -F(\hat{\mathbf{z}})$, we have

$$(J(\hat{\mathbf{z}}) + \lambda I)\nabla_\lambda \Delta\mathbf{z}(\lambda) + \Delta\mathbf{z}(\lambda) = \mathbf{0}, \quad (J(\hat{\mathbf{z}}) + \lambda I)\nabla_{\lambda\lambda}^2 \Delta\mathbf{z}(\lambda) + 2\nabla_\lambda \Delta\mathbf{z}(\lambda) = \mathbf{0}.$$

Rearranging the first equation implies that

$$\nabla_\lambda \Delta\mathbf{z}(\lambda) = -(J(\hat{\mathbf{z}}) + \lambda I)^{-1}\Delta\mathbf{z}(\lambda).$$

Combining the second equation with the expression of $\nabla_\lambda \Delta\mathbf{z}(\lambda)$ yields

$$\nabla_{\lambda\lambda}^2 \Delta\mathbf{z}(\lambda) = -2(J(\hat{\mathbf{z}}) + \lambda I)^{-2}\Delta\mathbf{z}(\lambda).$$

Putting these pieces together yields the desired expressions of $\psi'(\lambda)$ and $\psi''(\lambda)$.

It remains to show that $\psi(\lambda)$ is strictly decreasing on $(0, \infty)$. For $0 < \lambda_1 < \lambda_2$, we have

$$(J(\hat{\mathbf{z}}) + \lambda_1 I)\Delta\mathbf{z}(\lambda_1) = (J(\hat{\mathbf{z}}) + \lambda_2 I)\Delta\mathbf{z}(\lambda_2) = -F(\hat{\mathbf{z}}).$$

This implies

$$J(\hat{\mathbf{z}})(\Delta\mathbf{z}(\lambda_1) - \Delta\mathbf{z}(\lambda_2)) + \lambda_1 \Delta\mathbf{z}(\lambda_1) - \lambda_2 \Delta\mathbf{z}(\lambda_2) = \mathbf{0}.$$

Taking the inner product with $\Delta\mathbf{z}(\lambda_1) - \Delta\mathbf{z}(\lambda_2)$ yields

$$(\Delta\mathbf{z}(\lambda_1) - \Delta\mathbf{z}(\lambda_2))^\top J(\hat{\mathbf{z}})(\Delta\mathbf{z}(\lambda_1) - \Delta\mathbf{z}(\lambda_2)) + \lambda_1\|\Delta\mathbf{z}(\lambda_1)\|^2 - (\lambda_1 + \lambda_2)\Delta\mathbf{z}(\lambda_1)^\top\Delta\mathbf{z}(\lambda_2) + \lambda_2\|\Delta\mathbf{z}(\lambda_2)\|^2 = 0.$$

Since $v^\top J(\hat{\mathbf{z}})v \geq 0$ for all $v$, we have

$$\lambda_1\|\Delta\mathbf{z}(\lambda_1)\|^2 - (\lambda_1 + \lambda_2)\Delta\mathbf{z}(\lambda_1)^\top\Delta\mathbf{z}(\lambda_2) + \lambda_2\|\Delta\mathbf{z}(\lambda_2)\|^2 \leq 0.$$

This together with $\Delta\mathbf{z}(\lambda_1)^\top\Delta\mathbf{z}(\lambda_2) \leq \|\Delta\mathbf{z}(\lambda_1)\|\|\Delta\mathbf{z}(\lambda_2)\|$ and $\lambda_2 > \lambda_1$ yields that $\|\Delta\mathbf{z}(\lambda_1)\| > \|\Delta\mathbf{z}(\lambda_2)\|$ whenever $F(\hat{\mathbf{z}}) \neq \mathbf{0}$. This yields the desired result. $\qquad\square$

**Proof of Proposition 4.5.** Using Eq. (4.6), we have

$$\phi'(\lambda) = \tfrac{1}{2}\tfrac{\psi'(\lambda)}{\sqrt{\psi(\lambda)}} - \tfrac{1}{6\rho}.$$

By Proposition 4.4, $\psi'(\lambda) < 0$ for all $\lambda > 0$. This implies $\phi'(\lambda) < 0$. Since $\phi$ is continuous and strictly decreasing, the root is unique. The existence of such root follows from $\phi(\lambda) \to -\infty$ as $\lambda \to +\infty$ and the fact that $\phi(\lambda) > 0$ if $\lambda$ is sufficiently close to 0. $\qquad\square$

**Proof of Theorem 4.6.** By Proposition 4.5, $\phi$ is continuous and strictly decreasing. Thus, the bracketing interval remains valid throughout the iterations and contains $\lambda^\star$. Whenever the iterate $\tilde{\lambda}^{j+1}$ leaves the bracket, the method takes a bisection step, which halves the bracket length. After at most $O(\log((U^0 - L^0)/r))$ safeguarding steps, the bracket width is at most $r$, and the iterate satisfies $|\lambda^j - \lambda^\star| \leq r$.

By Proposition 4.5, we have $\phi(\lambda^\star) > 0$ and $\phi'$ is Lipschitz around the unique root $\lambda^\star$. Standard one-dimensional Newton analysis implies that there exists $r > 0$ such that if $|\lambda^j - \lambda^\star| \leq r$, the Newton step stays within the neighborhood, is accepted, and the iterates converge $Q$-quadratically. The number of Newton iterations required to reach the tolerance $\epsilon$ is thus $O(\log\log(1/\epsilon))$. Combining the safeguarding phase and the local Newton phase yields the desired result. $\square$

**Proof Proposition 4.7.** Since $S \succeq 0$, the map $\eta \mapsto (\eta I + S)^{-1}$ is operator monotone decreasing and operator convex on $(0, \infty)$, which implies that $\Psi(\eta)$ is strictly decreasing and convex. Subtracting the linear term $\eta/(36\rho^2)$ preserves convexity and strict monotonicity. This yields the desired result. $\square$

**Proof of Theorem 4.8.** By Proposition 4.7, we have $\Phi'(\eta^0) < 0$, and it follows from the update formula that $\eta^1 > \eta^0 > 0$. Proposition 4.7 also guarantees the convexity and differentiability of $\phi$ which together with the update formula implies that

$$\Phi(\eta^1) \geq \phi(\eta^0) + (\eta^1 - \eta^0)\Phi'(\eta^0) = 0.$$

Repeating this argument yields that $\eta^j > 0$ and $\Phi(\eta^j) \geq 0$. In addition, $\Phi$ is strictly decreasing. Thus, the iterates converge monotonically towards the unique solution. Suppose that $\eta^\star$ is the unique solution. Then, the mean value theorem implies

$$\Phi(\eta^j) = \Phi(\eta^\star) + (\eta^j - \eta^\star)\Phi'(\tilde{\eta}) = (\eta^j - \eta^\star)\Phi'(\tilde{\eta}), \quad \text{for some } \tilde{\eta} \in (\eta^j, \eta^\star).$$

Combining this with the update formula yields

$$|\eta^\star - \eta^{j+1}| = \left| (\eta^\star - \eta^j)\left(1 - \frac{\Phi'(\tilde{\eta})}{\Phi'(\eta^j)}\right) \right| \leq |\eta^\star - \eta^j| \cdot \left| 1 - \frac{\Phi'(\tilde{\eta})}{\Phi'(\eta^j)} \right|.$$

Since $\Phi$ is convex, we have $\Phi'(\lambda^0) \leq \Phi'(\lambda^j) \leq \Phi'(\tilde{\lambda}) \leq \Phi'(\lambda^\star) < 0$ which implies that

$$0 \leq 1 - \frac{\Phi'(\tilde{\eta})}{\Phi'(\eta^j)} \leq 1 - \frac{\Phi'(\eta^\star)}{\Phi'(\eta^0)} < 1.$$

Putting everything together then implies that the convergence rate is globally $Q$-linear with a factor at least $1 - \Phi'(\eta^\star)/\Phi'(\eta^0)$. The asymptotic $Q$-quadratic convergence of the Newton iteration follows. This together with $\lambda = \sqrt{\eta}$ yields the desired result. $\square$

**Proof of Lemma 4.11.** Fixing $\mathbf{z} \in \mathbb{R}^{m+n}$, we obtain from Eq. (4.8) and (4.10) that $J(\mathbf{z}) = \frac{1}{|\mathcal{S}|}\sum_{j=1}^{|\mathcal{S}|} J_j$ where each random matrix $J_j$ is random and satisfies that $\text{Prob}(J_j = \frac{1}{p_i}\Lambda_i \Sigma_{ii} \Lambda_i^\top) = p_i$ with $p_i > 0$ in Eq. (4.12). For simplicity, we define

$$X_j = J_j - DF(\mathbf{z}) = J_j - \Lambda^\top \Sigma \Lambda, \qquad X = \sum_{j=1}^{|\mathcal{S}|} X_j = |\mathcal{S}|(J(\mathbf{z}) - \Lambda^\top \Sigma \Lambda).$$

It is easy to verify that $\mathbb{E}[X_j] = 0$ and

$$\|\mathbb{E}[X_j^2]\| \leq \left( \frac{1}{N}\sum_{i=1}^{N} \|DF_i(\mathbf{a}_i^\top \mathbf{x}, \mathbf{b}_i^\top \mathbf{y})\|(\|\mathbf{a}_i\|^2 + \|\mathbf{b}_i\|^2) \right)^2 \overset{\text{Eq. (4.11)}}{\leq} B_{\text{avg}}^2.$$

Applying the operator-Bernstein inequality yields

$$\text{Prob}(\|J(\mathbf{z}) - DF(\mathbf{z})\| \geq \tau) = \text{Prob}(\|X\| \geq \tau|\mathcal{S}|) \leq 2(m+n)\exp\left( \frac{\tau^2 |\mathcal{S}|}{4B_{\text{avg}}^2} \right) \leq \delta.$$

This completes the proof. $\square$

**Proof of Theorem 4.13.** Since Algorithm 3 is a combination of Algorithm 2 and the random sampling strategy in Eq. (4.8) with $0 < \tau_k \leq \min\{\tau_0, \frac{\rho(1-\kappa_m)}{4(B_{\max}+6\rho)}\|F(\hat{\mathbf{z}}_k)\|\}$ and $\kappa_H = B_{\max}$, we can obtain the desired results from Theorems 4.1 and 4.6 if the following statement holds true:

$$\text{Prob}(\|J(\hat{\mathbf{z}}_k) - DF(\hat{\mathbf{z}}_k)\| \leq \tau_k \text{ for all } 0 \leq k \leq T-1) \geq 1 - \delta. \tag{D.6}$$

To guarantee an overall accumulative success probability of $1 - \delta$ across all $T$ iterations, it suffices to set the per-iteration failure probability as $1 - \sqrt[T]{1-\delta}$ as we have done in Algorithm 3. Moreover, we have $1 - \sqrt[T]{1-\delta} = O(\frac{\delta}{T}) = O(\delta\epsilon^{2/3})$. Since this failure probability has only been proven to appear in the logarithmic factor for the sample size in both Lemma 4.9 and 4.11, the extra cost will not be dominating. As such, when Algorithm 3 terminates, all of the Jacobian approximations have satisfied Eq. (D.6). This completes the proof. □

