# OpenReview forum: "Explicit Second-Order Min-Max Optimization: Practical Algorithms and Complexity Analysis"
_TMLR — Accepted by TMLR_

### Review · Reviewer_MfyV · 2026-02-20

**Summary Of Contributions:**

This paper studies several second-order algorithms for solving convex-concave unconstrained minimax optimization. The proposed inexact methods achieves convergence rate of $O(\epsilon^{-2/3})$. The authors also provide a simple method for solving the subproblem, which requires $O(\log\log(1/\epsilon))$  calls to the lienar system solver.

**Audience:**

Yes

**Audience Explanation:**

Solving convex-concave minimax optimization is an important problem in the field of machine learning. I believe researchers in the optimization community will be interested in the findings of this paper.

**Broader Impact Concerns:**

no concerns

**Claims And Evidence:**

Yes

**Claims Explanation:**

Though I didn't check the proofs in the appendix, the theoretical results appear to be reasonable. Experimental results verify the effectiveness of the proposed methods.

**Requested Changes:**

Since the iteration complexity of the proposed methods is worse than the best-known result of $O(\epsilon^{-4/7})$, the paper should demonstrate that the proposed methods are more practical in real-world scenarios. Therefore, I believe it would be valuable for the paper to include experiments on real-world applications to showcase their superior performance. Also, a discussion on the sensitivity of the hyperparameters would be beneficial.

---

> ### Author Response · Authors · 2026-03-03
> **Response to Reviewer MfyV**
>
> Thank you for the constructive suggestions. We have revised the manuscript accordingly.
>
> **Real-world experiments**. We added a real-data evaluation on AUC maximization, using three LIBSVM datasets (a9a, covtype, w8a). We report the restricted-gap metric versus both epoch count and wall-clock time (see Figure 2), complementing the synthetic results and demonstrating stable behavior on practical finite-sum problems.
>
> **Hyperparameter sensitivity**. We added a dedicated discussion clarifying the main user-facing parameters in the algorithm: (i) the safeguard parameter $\kappa_m$ in the inexact-solve condition, (ii) the subsampling parameter $\delta$ (affecting $|S_k|$ only through logarithmic factors), and (iii) numerical tolerances in the one-dimensional root-finding routine used to compute the step parameter. We summarize the qualitative tradeoffs (e.g., tighter tolerances or smaller $\kappa_m$ may increase the number of linear solves per iteration but can improve accuracy), and we emphasize that our experiments use a single default configuration without dataset-specific tuning, which already yields consistent behavior across datasets.
>
> We hope these revisions address the reviewer’s concerns and improve clarity, positioning, and practical guidance.

---

### Review · Reviewer_gHHn · 2026-02-22

**Summary Of Contributions:**

This work studied unconstrained convex-concave min-max optimization problems, it proposed a conceptual second-order algorithmic framework, then proposed several inexact regularized Newton-type algorithms based on the framework including finite-sum problems. The subproblem solver design is discussed, the iteration complexities and computation complexities are provided. Numerical experiment results are provided to verify the effectiveness of the proposed algorithms.

**Audience:**

Yes

**Audience Explanation:**

Min-max optimization is an important topic with many useful applications in ML, the development of second-order method is attractive as an appealing alternative to first-order methods. So the topic of the work should be interesting to many TMLR's readers.

Regarding the results of the work, even though the theoretical convergence rate is not SOTA compared to other recent works, the proposed algorithm is relatively elegant and easy to implement, also the experiment results reveal the potential of proposed algorithms.

**Broader Impact Concerns:**

This is a purely theoretical work, no ethical issues here.

**Claims And Evidence:**

Yes

**Claims Explanation:**

This work proposed several second-order algorithms for unconstrained convex-concave min-max optimization problems, the algorithm design is based on Newton's method and Extragradient, which is convincing in min-max optimization; also the theoretical convergence guarantees are provided with proof supported; the numerical experiments further validify the effectiveness of proposed algorithms.

**Requested Changes:**

1. You used the restricted gap function as the measurement, can you explain how to calculate it in your practical experiments, are there closed-form solutions for those problems?
2. Is there any experiment results in the high-dimensional case? Currently the experiment results seems to outperform in relatively small-scale problems, and the gap of outperformance seems to squeeze as the dimension becomes larger.
3. For Section 4.2, currently I have not got how the solver achieves Condition 4.1 and 4.2, I understand that the convergence should be fast and the two conditions can be satisfied, but how does $\lambda^j\rightarrow\lambda^*$ transfer to Condition 4.1 and 4.2, also how the correlated complexities changes with respect to the parameters in Condition 4.1 and 4.2? Can you further add the details?

---

> ### Author Response · Authors · 2026-03-04
> **Response to Reviewer gHHn**
>
> Thank you for the constructive suggestions. We have revised the manuscript accordingly.
>
> **Restricted gap computation.** We added Remark 2.7 to clarify how the restricted gap is computed in practice. When a closed-form saddle point $\textbf{z}^\star$ is available (cubic regularized bilinear benchmark), we compute the restricted gap exactly; when $\textbf{z}^\star$ is unavailable (AUC), we evaluate the restricted gap relative to a high-accuracy proxy (best-found solution under a long budget with a stringent stopping rule).
>
> **Higher-dimensional experiments.** We added high-dimensional results up to $n=1000$ and report both iteration counts and wall-clock time (Figure 1). We also discuss that relative margins may narrow at larger dimensions when conditioning and linear-algebra costs dominate, while the method remains stable in time-to-accuracy.
>
> **Section 4.2 clarifications (Conditions 4.1-4.2; $\lambda^j \to \lambda^\star$ transfer; parameter dependence).** We clarified the modular roles of the two conditions: Condition 4.1 is enforced by Jacobian construction (Section 4.3), while Section 4.2 focuses on the inner subproblem for fixed $J(\hat{\textbf{z}})$. Crucially, we provide an explicit link between the scalar residual $|\phi(\lambda)|$ and the Condition 4.2 residual, yielding a verifiable stopping rule; thus $\lambda^j \to \lambda^\star$ implies Condition 4.2 after finitely many inner iterations, whereas Condition 4.1 is independent of $\{\lambda^j\}$. We also state the dependence of the admissible inexactness level $\tau_k$ on $(\kappa_m,\kappa_J,\rho)$ and $\|F(\hat{\textbf{z}}_k)\|$ in Theorem 4.1.
>
> We hope these revisions address the reviewer’s concerns and improve clarity, positioning, and practical guidance.

---

### Review · Reviewer_eQfq · 2026-02-26

**Summary Of Contributions:**

This paper proposes a class of explicit second-order methods for solving unconstrained convex--concave min--max optimization problems. In this paper, the authors examine how second-order information is used to speed up extra-gradient methods, even under inexactness. The authors proposed a conceptual Newton-type extra-gradient framework achieving an iteration complexity of $\epsilon^{-2/3}$ in terms of a restricted gap function. The authors also propose one inexact second-order variant that allows inexact Jacobian information, inexact subproblem solutions, and subsampled Jacobians for finite-sum problems. The paper also include a safeguarded Newton--bisection subroutine for solving the cubic-regularized subproblem using single Schur decomposition and requires $O(\log\log(1/\epsilon))$ to solve the linear system. The total complexity bound of the proposed algorithms is $O\left((m+n)^\omega \epsilon^{-2/3} + (m+n)^2 \epsilon^{-2/3}\log\log(1/\epsilon)\right)$. The paper positions itself as improving line-search-based second-order min-max methods by removing implicit search and improving the $\log(1/\epsilon)$ dependence to $\log\log(1/\epsilon)$. The authors also conduct numerical experiments on synthetic and real data to demonstrate the efficiency of the proposed methods.

**Audience:**

Yes

**Audience Explanation:**

The topic is highly relevant to the optimization and machine learning community. Saddle-point problems remain central in adversarial learning and game-theoretic ML. The paper is closely related to the interesting topic include inexact second-order oracle models, proof of lower bounds under Jacobian noise, and the explicit versus implicit second-order updates using extra-gradients.

**Broader Impact Concerns:**

There is no Broader Impact Concerns for this paper.

**Claims And Evidence:**

Yes

**Claims Explanation:**

The authors present clear problem focus and motivation in this submission. The paper addresses a concrete and well-motivated question: Can explicit second-order min-max methods achieve global complexity guarantees under inexact second-order information? This question is timely and relevant, especially given the importance of saddle-point problems in modern machine learning (e.g., GANs, adversarial training, and multi-agent systems). The authors propose the inexact approach to solve the problem. The major strength is the rigorous formulation of the inexact Jacobian regularity conditions and the sufficient inexact subproblem solving conditions. The analysis carefully quantifies allowable Jacobian error and provides high-probability guarantees for subsampled variants. This level of rigor is stronger than many existing second-order min-max works.

The subproblem solver analysis is non-trivial. The scalar reformulation of the cubic regularized subproblem and the safeguarded Newton--bisection scheme are technically well developed. The distinction between general asymmetric Jacobian (requiring safeguarding), and the antisymmetric Jacobian (globally convergent pure Newton), is clearly explained and mathematically sound. The $O(\log\log(1/\epsilon))$ phase is carefully justified. The method matches the best-known iteration complexity $O(\epsilon^{-2/3})$ while allowing inexact second-order information. This extension is nontrivial, as many lower bounds assume exact oracle access. This is the major theoretical contributions of this submission and all theoretical results are provided with rigorous mathematical proof.

The authors also provided empirical evidence. The numerical experiments demonstrate clear superiority of second-order methods over first-order ones, competitive performance relative to recent second-order line-search methods, and improved iteration counts and runtime in several settings. These empirical results from the numerical experiments section are consistent with the theoretical analysis.

**Requested Changes:**

Here are the weakness of this submission that the authors should improve in the revised version of the paper.

All the algorithms need to know the values of the Hessian smoothness parameter $\rho$. This Hessian smoothness parameter can be difficult to be estimated in practice. Hence, the proposed algorithms are not parameter-free and the implementation of these algorithms depend on the estimation of the Hessian smoothness parameter $\rho$. This weaken the practical application of the proposed algorithms.

There is no strong conceptual novelty from this submission. The framework builds heavily on some prior related works (e.g., Lin & Jordan; Adil et al.; Xu et al.; Cartis et al.). The main novelty lies in combining components and improving logarithmic factors. While technically nontrivial, the conceptual advancement is incremental rather than foundational.

The paper does not establish optimality under the proposed inexact oracle model. Although lower bounds under exact oracles are discussed, the gap under inexact Jacobian models remains open. A clearer positioning relative to recent oracle-complexity results would strengthen the contribution.

The complexity includes a term of order $(m+n)^\omega$, which may limit practical applicability for very large-scale problems. A deeper discussion of memory cost and large-scale feasibility would be beneficial.

The experiments focus on a cubic-regularized bilinear synthetic problem. The experiments didn't include real-world machine learning benchmarks using non-quadratic or linear objective functions, finite-sum subsampled experiments, sensitivity analysis to Jacobian noise, and the comparison with other algorithms such as the quasi-Newton methods. Given the ``practical'' emphasis of the paper, broader empirical validation would improve the impact.

---

> ### Author Response · Authors · 2026-03-04
> **Response to Reviewer eQfq**
>
> Thank you for the constructive suggestions. We have revised the manuscript accordingly.
>
> **Dependence on Hessian smoothness/not parameter-free.** We agree that having an exact Hessian-Lipschitz constant is generally unrealistic. Our analysis uses a constant $\rho$ to state sufficient conditions and keep the complexity statements clean, but the algorithm can be implemented without knowing $\rho$ a priori via a standard backtracking mechanism that increases the regularization parameter until the acceptance/decrease condition holds. We have added an explicit discussion of this issue and the practical remedy (Remark 4.3). We also clarify that a fully "parameter-free" variant (automatically calibrating all schedules) is an interesting direction for future work.
>
> **Conceptual novelty/combining components.** We appreciate this concern and have revised the introduction to sharpen the conceptual contribution and clarify how it differs from prior second-order min-max methods. While our approach builds on ideas from cubic regularization, inexact Newton methods, and safeguarded globalization, the key conceptual step is to explicitize and modularize the interaction between two practical error sources that typically coexist: (i) inexact construction of second-order information (e.g., via finite-sum subsampling), and (ii) inexact subproblem solves. Existing works commonly assume one of these is exact; our framework quantifies their joint effect and yields a concrete safeguarded Newton-type method that provably terminates with controlled inner work per iteration (including the $\log\log(1/\epsilon)$ behavior in the safeguarded solver).
> We have rewritten the ``Contributions'' paragraph accordingly.
>
> **No optimality claim under the inexact oracle model.** We agree and do not claim information-theoretic optimality under our inexactness model. We added an explicit remark (see Remark 4.2) clarifying why existing lower bounds (typically stated for exact oracles and/or exact inner solves) do not directly apply, and we explain how oracle definitions change when second-order information is constructed approximately with a decaying-accuracy schedule. We view establishing tight lower bounds for this inexact model as an important open direction.
>
> **Per-iteration cost $(m+n)^\omega$ and scalability; memory requirements.** We agree that the $(m+n)^\omega$ per-iteration factor limits direct applicability in extremely large-scale settings. We have added a clear discussion in Section 4.2: the factor stems from a one-time Schur decomposition to exploit quasi-upper-triangular structure, and the method requires $O((m+n)^2)$ memory to store the factors. We also clarify that our method is best suited to moderate dimensions/structured problems and outline practical variants (iterative Krylov solvers with preconditioning, randomized/truncated decompositions, and block-structured updates) as future directions for large-scale regimes.
>
> **Experiments too narrow; request broader ML benchmarks, Jacobian-noise sensitivity, quasi-Newton comparisons.** We agree that a comprehensive empirical evaluation would include broader benchmarks, systematic sensitivity studies, and quasi-Newton/limited-memory baselines. In this revision, we expand beyond synthetic instances by adding AUC maximization on standard LIBSVM datasets, which is a representative non-quadratic finite-sum problem and directly matches the subsampling/inexact-Jacobian regime analyzed in the paper. We evaluate the subsampled variant that constructs $J(\hat{\textbf{z}_k)$ from minibatches (see Algorithm 3). We also clarified the scope of the empirical study: in finite-sum settings the minibatch size and accuracy schedule provide a principled control knob for subsampling-induced "Jacobian noise", and our safeguarded globalization is designed to remain stable under moderate inaccuracies while tightening accuracy as higher precision is required. At the same time, we avoid over-claiming: we present the results as evidence of stable performance on the tested instances rather than a universal dominance claim. We view broader benchmark suites, quasi-Newton comparisons, and dedicated noise-sensitivity sweeps as important future work (now stated explicitly in the experiments discussion).
>
> We hope these revisions address the reviewer’s concerns and improve clarity, positioning, and practical guidance.

---

### Author Response · Authors · 2026-03-04
**Author response submitted & updated manuscript**

We thank all the reviewers for your time and for the constructive, insightful feedback. We have responded to all questions and updated the manuscript; non-trivial changes are highlighted in red.

Best Regards,

Authors of Submission 7054

---

### Decision · Action_Editor_WJBS · 2026-05-23

**Recommendation:** Accept as is

**Audience:**

Yes

**Audience Explanation:**

Unconstrained convex-concave min-max optimization is a highly relevant topic within the machine learning community, particularly for its applications in adversarial training, multi-agent systems, and GANs. It is also a module that appears also in nonconvex min-max optimization. The development of explicit second-order methods with rigorous analysis is undoubtedly of interest to many researchers in the community.

**Claims And Evidence:**

Yes

**Claims Explanation:**

The reviewers reached a strong consensus that the theoretical formulations are rigorous and well-supported. Furthermore, the authors successfully addressed the reviewers' initial concerns regarding practical applicability by adding real-world experiments. These additions provided convincing empirical evidence that aligns well with the theoretical claims

---

> ### Author Response · Authors · 2026-05-26
> **Camera-Ready Submission and Code Link for Paper 7054**
>
> Dear AE,
>
> Thank you very much for handling our paper and for all your time and effort throughout the process. We have uploaded the camera-ready version and also included the link to our code.
>
> Please let us know if there are any further questions or comments. We truly appreciate all your help and support.
>
> Sincerely,
>
> Authors of Paper 7054

---

> > ### Comment · Action_Editor_WJBS · 2026-05-26
> >
> > Thank you! I confirmed the submission.